



# Effect of Disorder on Bulk Sound Wave Speed : A Multiscale Spectral Analysis.

Rohit Kumar Shrivastava[1] and Stefan Luding[1]

[1]Multiscale Mechanics (MSM), MESA+, Faculty of Engineering Technology (CTW), PO Box 217, 7500 AE Enschede, Netherlands

*Correspondence to:* Rohit Kumar Shrivastava(r.k.shrivastava@utwente.nl)

**Abstract.** Disorder of size (polydispersity) and mass of discrete elements/particles in randomly structured media (e.g. granular matter like soil) has numerous effects on the materials' sound propagation characteristics [Mouraille and Luding, Ultrasonics, 48, 498-505 (2008); Lawney and Luding, Acta Mechanica, 225, 2385-2407 (2014)]. The influence of disorder on the sound wave speed and its low pass frequency filtering characteristics is the subject of this study. Goal is understanding the connection between the particle-micro-scale disorder and dynamics and the system-macro-scale wave propagation which can be applied to non-destructive testing, seismic exploration of buried objects (oil, mineral, etc.) or to study the internal structure of earth. To isolate the longitudinal P-wave mode from shear and rotational modes, a one-dimensional system of elements/particles is used to study the effect of mass disorder alone via ensemble averaged real time signals, signals in Fourier space and dispersion curves. Increase in polydispersity can decrease the sound wave speed because of a decrease in the number of contacts between particles [Mouraille and Luding, Ultrasonics, 48, 498-505 (2008)]. Also, increase in mass disorder (where disorder has been defined such that it is independent of the shape of the probability distribution of masses) decreases the sound wave speed along a particular granular chain. Energies associated with the eigenmodes are constant, independent of time and have been used to derive dispersion relations for disordered chains, these dispersion relations show a decrease in cut-off frequency and thus wave speed with increasing disorder.

## 1 Introduction

Sound wave propagation through matter has been an extensive area of research (as textbook example, see Aki and Richards (2002)) whether it may be applied for the study of earthquakes, internal structure of earth, oil, gas or mineral exploration (seismology), dissecting human body without using blades, revealing material properties through non-destructive testing (ultrasonics), studying the structure of lattices or for designing metamaterials. There are numerous applications and uncountable problems which still need to be solved. However, the challenge has always been to resolve into the finest structures of any media using wave propagation and hence, steps are being taken in the direction of microseismics, see e.g. O'Donovan et al. (2016).

Disordered/Heterogeneous/Random media cause multiple scattering of seismic waves, which eventually causes them to become dispersed, attenuated and localized in space (Sato (2011), Scales and Van Vleck (1997)). This phenomena of multiple



scattering causes the formation of the so called coda which is the tail part of a propagating wave, while coda was earlier treated as noise (Weaver (2005)), now it has given way to coda wave interferometry with multiple applications (Snieder et al. (2002)), they have been studied in detail in laboratory experiments with uniaxial or triaxial apparatus for e.g. pulse propagation across glass beads (Jia et al. (1999)), indicating extreme sensitivity towards system configuration and getting washed out on ensemble

averaging with only the coherent part of the signal remaining. In Van Der Baan (2001), it was shown that macroscopic/seismic waves governed by the classical wave equation did not exhibit localization at lower frequencies but, this idea got repudiated by Larose et al. (2004), where weak localization (a mesoscopic phenomenon, precursor to wave localization; Sheng (2006)) was experimentally observed at a frequency as low as 20 Hz, indicating the inadequacy of the classical wave equation.

In recent years, wave propagation through granular materials has attracted a lot of attention. Granular material is a hetero-
geneous media with many discretized units and can be used for modeling geometrically heterogeneous media (Matsuyama and Katsuragi (2014)). The work done in ordered/disordered lattices for wave propagation (Gilles and Coste (2003), Coste and Gilles (2008), etc.) has also been used to understand wave propagation in granular materials through dispersion relations, frequency filtering, etc. For granular matter there are scaling laws which can relate its various physical parameters like density, pressure, coordination number, etc., forming an Effective Medium Theory (EMT) for granular matter (Makse et al. (2004)).

Nesterenko (1983) showed the existence of localized wave packets propagating in a non-linear granular chain (one dimen-sional granular material) under the condition of "sonic vacuum" (in the limit of zero acoustic wave speed and vanishing con-fining pressure) thus forming supersonic solitary waves or shocks, such concepts have been exploited immensely to develop various kinds of metamaterials like for shock and energy trapping (Daraio et al. (2006)), an acoustic diode (Boechler et al. (2011)) or for understanding and studying jamming transitions in granular matter (van den Wildenberg et al. (2013), Upad-
hyaya et al. (2014)). Some of the open questions and developments related to wave propagation in unconsolidated granular matter like higher harmonics generation, non-linear multiple scattering, soft modes, rotational modes, etc. have been addressed in Tournat and Gusev (2010). However, the focus of attention will not be on solitons and unconsolidated granular matter, hence, there will be no occurrence of sonic vacuum during analyses (no opening and closing of contacts of particles).

Another striking characteristic of a consolidated granular matter is that the grain-grain forces are correlated in a line like
manner and are known as force chains (Somfai et al. (2005)) which carry the large forces of the system and supposedly support faster sound transmission across granular matter (Ostojic et al. (2006)). In Owens and Daniels (2011), it was seen from experiments with 2 dimensional photo elastic disks that vibration propagates along the granular chains depicted by the brightness due to compression between the particles, however, the exact mechanism of propagation of the vibration is still a matter of ongoing research. Our system of investigation will be a single one of such granular chains (Fig. 2), it will assist in
isolating the P-wave or the longitudinal excitation from all other kinds of excitations (S-wave, rotational wave, etc). In Merkel et al. (2010) it was seen that inclusion or removal of rotation does not affect the longitudinal mode in an ordered granular crystal. However, the situations when rotations become prominent and cannot be ignored can be noted from Yang and Sutton (2015), see also Merkel and Luding (2016) and the references therein. Even though very simplistic, a polydisperse granular chain can have two kinds of disorder, mass disorder and stiffness disorder. However, the mass disorder has much stronger



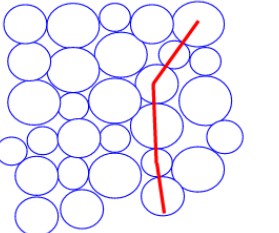 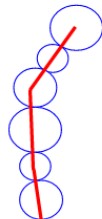

**Figure 1.** A granular/force chain from a network (schematic).

contribution towards disorder than stiffness because mass $\propto$ radius$^3$ whereas, stiffness $\propto$ radius$^{1/3}$ (Achilleos et al. (2016)). Only mass disorder for the disordered granular chain has been chosen.

In Sect. 2 an impulse propagating across a granular chain has been modeled. A similar model was used in Marketos and O'Sullivan (2013) and Lawney and Luding (2014). Section 3 mentions the dispersion relation for wave propagation across a granular media, group velocity and proposes a novel way of computing the dispersion relation in terms of moment of eigen-modal energy. In Sect. 4, the equations mentioned in Sect. 2 and 3 have been computed numerically and the observations have been discussed. Finally Sect. 5 concludes the observations made in Sect. 4 with Sect. 2 and 3 as the foundations. An outlook of the ongoing as well as possible future research work on wave propagation in granular matter has also been mentioned.

## 2  Modeling a general one-dimensional chain

A one-dimensional chain of $N+2$ particles has been taken into consideration. Each particle $i$ has mass $\tilde{m}_i$ and contact stiffness $\tilde{\kappa}_{(i,j)}$ with respect to a neighboring particle $j$. The tilde symbols have been used for dimensional quantities. The interaction force experienced by neighboring particles $i$ and $j$ is :

$$\tilde{F}_{(i,j)} = \tilde{\kappa}_{(i,j)}\tilde{\delta}_{(i,j)}^{1+\beta}, \qquad \tilde{\delta}_{(i,j)} \geq 0, \tag{1}$$

with the contact stiffness $\tilde{\kappa}_{(i,j)}$ dependent on the value of $\beta$, which changes with change in properties of the contacting bodies and the particle overlap $\tilde{\delta}_{(i,j)} = \tilde{r}^{(i)} + \tilde{r}^{(j)} - |\tilde{x}^{(j)} - \tilde{x}^{(i)}|$, with the radius $\tilde{r}$ and co-ordinates $\tilde{x}$ of the centers of particles. The Hertzian and linear model are given by $\beta = 1/2$ and $\beta = 0$, respectively (Lawney and Luding (2014)). In the framework of the Discrete Element Method, the overlap of particles substitutes their deformations at the contacts, which would be much more difficult to resolve.

### 2.1  Non-dimensionalization

A length scale $\tilde{\ell}$ is used such that the scaled particle overlap $\delta_{(i,j)} = \tilde{\delta}_{(i,j)}/\tilde{\ell}$ yields,

$$\left|\tilde{F}_{(i,j)}\right| = \tilde{\kappa}_{(i,j)}\tilde{\ell}^{1+\beta}\delta_{(i,j)}^{1+\beta}, \tag{2}$$





At rest :

**Figure 2.** Chain of granular elements during dynamic wave propagation with length scaled by the characteristic equilibrium distance $\tilde{\Delta}_o$

If the chain is compressed by an applied force $\tilde{F}_o$, the initial dimensionless inter-particle overlap between the particles $i$ and $j$ is,

$$\Delta_{(i,j)} = \left( \frac{\tilde{F}_o}{\tilde{\kappa}_{(i,j)} \tilde{\ell}^{1+\beta}} \right)^{1/(1+\beta)} . \tag{3}$$

There are several length scales $\tilde{\ell}$ that could be chosen, e.g. the particle size, the driving amplitude or the characteristic overlap of

5  the particles in static equilibrium. The latter has been chosen for computations here. The characteristic overlap of the particles in static equilibrium ($\tilde{\Delta}_o$) is obtained when all the contact stiffness ($\tilde{\kappa}_{(i,j)}$) of particles are chosen as $\tilde{\kappa}_o$, which is defined as the characteristic contact stiffness,

$$\tilde{\ell} = \tilde{\Delta}_o = \left( \frac{\tilde{F}_o}{\tilde{\kappa}_o} \right)^{1/(1+\beta)}, \tag{4}$$

Other dimensionless quantities are, the mass $b = \tilde{m}/\tilde{M}_1$ where, $\tilde{M}_1$ is the first moment of the mass distribution of the particles

10  of the media as shown in Appendix B (which is also the unscaled average mass of the particles of the media), the dimensionless displacement $u = \tilde{u}/\tilde{l}$ and the dimensionless spring constant $\kappa = \tilde{\kappa}/\tilde{\kappa}_o$, the characteristic time scale becomes

$$\tilde{t}_c = \frac{1}{\tilde{\ell}^{\beta/2}} \sqrt{\frac{\tilde{M}_1}{\tilde{\kappa}_o}}, \tag{5}$$





which gives us the dimensionless time $\tau = \tilde{t}/\tilde{t}_c$. The displacement of particle $i$ from its equilibrium position $\tilde{x}_o^{(i)}$ is $\tilde{u}^{(i)} = \tilde{\ell}u^{(l)} = \tilde{x}^{(i)} - \tilde{x}_o^{(i)}$, so that the overlap becomes, $\delta_{(i,j)} = \Delta_{(i,j)} - (u^{(j)} - u^{(i)})$. Finally, the interaction force scales as,

$$\tilde{F}_{(i,j)} = \frac{\tilde{M}_1 \tilde{\ell}}{\tilde{t}_c^2} F_{(i,j)}. \tag{6}$$

## 2.2 Equations of Motion and Linearization

The equation of motion for any particle $i$ (except the boundary particles at either end of the chain) by using Eq. (2) and (6), and non-dimensionalization as mentioned in Sect. 2.1, can be written as,

$$b^{(i)}\frac{d^2 u^{(i)}}{d\tau^2} = F_{(i-1,i)} + F_{(i,i+1)} = \kappa_{(i-1,i)}\delta_{(i-1,i)}^{1+\beta} - \kappa_{(i,i+1)}\delta_{(i,i+1)}^{1+\beta}, \tag{7}$$

and force can be expressed as a power series, expanded about the initial overlap $\Delta_{(i,j)}$ due to pre-compression,

$$F_{(i,j)} = \kappa_{(i,j)}\Delta_{(i,j)}^{1+\beta} + \kappa_{(i,j)}(1+\beta)\Delta_{(i,j)}^{\beta}(\delta_{(i,j)} - \Delta_{(i,j)}) + \frac{1}{2}\kappa_{(i,j)}\beta(1+\beta)\Delta_{(i,j)}^{\beta-1}(\delta_{(i,j)} - \Delta_{(i,j)})^2 + \dots$$

Under small displacements from equilibrium condition (during wave propagation), using $\delta_{(i,j)} = \Delta_{(i,j)} - (u^{(j)} - u^{(i)})$ and after ignoring higher order non-linear terms, we arrive at

$$F_{(i,j)} = \kappa_{(i,j)}\Delta_{(i,j)}^{1+\beta} - \kappa_{(i,j)}(1+\beta)\Delta_{(i,j)}^{\beta}\left(u^{(j)} - u^{(i)}\right). \tag{8}$$

Using the previous force relation in Eq. (7), we get the following equation of motion,

$$b^{(i)}\frac{d^2 u^{(i)}}{d\tau^2} = \kappa_{(i-1,i)}\Delta_{(i-1,i)}^{\beta}\left[\Delta_{(i-1,i)} - (1+\beta)(u^{(i)} - u^{(i-1)})\right] - \kappa_{(i+1,i)}\Delta_{(i,i+1)}^{\beta}\left[\Delta_{(i+1,i)} - (1+\beta)(u^{(i+1)} - u^{(i)})\right]. \tag{9}$$

With $\beta = 0$ the equation becomes linearized (Under high confining force the linear contact model is valid; In Sen and Sinkovits (1996) it was verified that a granular chain under high pre-confining force has the same frequency response to small amplitude signals as that of a harmonic chain),

$$b^{(i)}\frac{d^2 u^{(i)}}{d\tau^2} = \kappa_{(i-1,i)}\left[\Delta_{(i-1,i)} - (u^{(i)} - u^{(i-1)})\right] - \kappa_{(i+1,i)}\left[\Delta_{(i+1,i)} - (u^{(i+1)} - u^{(i)})\right]. \tag{10}$$

Since the focus of our study is not on the occurrence of sonic vacuum (Nesterenko (1983)), the initial impulse $(v_o)$ should be

chosen to avoid opening of contacts. Since, we are interested only in mass disorder, we can choose all coupling stiffness $(\kappa_{(i,j)})$ as 1, $\Delta_{(i,j)} = 1$ without loss of generality. Now, Eq. (10) for individual particles can be written as,

$$b^{(i)}\frac{d^2 u^{(i)}}{d\tau^2} = u^{(i+1)} - 2u^{(i)} + u^{(i-1)} \tag{11}$$

This results in $N$ equations which eventually can be expressed in the matrix form,

$$\mathbf{M}\frac{d^2\mathbf{u}}{d\tau^2} = \mathbf{K}\mathbf{u} + \mathbf{f}, \tag{12}$$

where $\mathbf{M}$ is a diagonal mass matrix with entries $b^{(1)}, b^{(2)}, b^{(3)}, ..., b^{(N)}$ and zero otherwise; $\mathbf{K}$ is a matrix with diagonal entries -2, superdiagonal and subdiagonal entries +1 and zero otherwise (for $\kappa = 1$). $\mathbf{f}$ is the external force which depends on the specified driving. Introducing, $\mathbf{A} = -\mathbf{M}^{-1}\mathbf{K}$ then, Eq. (12) can be written as,

$$-\frac{d^2\mathbf{u}}{d\tau^2} = \mathbf{A}\mathbf{u} - \mathbf{M}^{-1}\mathbf{f}. \tag{13}$$



### 2.3 Analysis in real space/spatial Fourier space :

Using an ansatz for real space and another ansatz for spatial Fourier space in Eq. (13) (the calligraphic fonts from now onwards will depict the spatial Fourier transform counterparts of the real space parameters),

$$\mathbf{u} = \mathbf{u}' e^{i\omega t} \quad \text{or} \quad \mathcal{U} = \mathcal{U}' e^{i(\omega t - ku)}, \tag{14}$$

one has

$$\mathbf{A}\mathbf{u} = \omega^2 \mathbf{u} \quad \text{or} \quad \mathbf{A}\mathcal{U} = \omega^2 \mathcal{U}, \tag{15}$$

where $k$ is the wavenumber and $\mathcal{U} = \int\limits_{-\infty}^{\infty} \int\limits_{-\infty}^{\infty} u e^{-i(\omega t - ku)} \mathrm{d}t\,\mathrm{d}u$ is the double Fourier transform (spatial as well as temporal) ansatz. Equation (15) is a familiar eigen value problem. The eigenvalues $\omega_j^2$ and eigenvectors $\mathbf{s}_{(j)}$ of the matrix $\mathbf{A}$ give the eigendomain of the granular chain that are independent of the external driving. The square root of the eigenvalues, $\omega_j$, are the natural frequencies of the chain. The set of eigenvectors can be orthonormalised to obey the orthonormality condition,

$$\mathbf{s}_{(i)}^{\mathrm{T}} \mathbf{M} \mathbf{s}_{(j)} = \delta_{ij}, \tag{16}$$

with $\delta_{ij}$ being the Kronecker delta symbol. The $\mathbf{S}$ matrix or the eigenbasis matrix can be constructed with $\mathbf{s}_{(j)}$ as the columns of the matrix, which can be used to transform back and forth from real domain to eigen domain. The columns ($\mathbf{s}_{(j)}$) of the matrix $\mathbf{S}$ are sorted such that the corresponding eigenvalues $\omega_j$ are in increasing order. The vector of eigenmode amplitudes is,

$$\mathbf{z} = \mathbf{S}^{-1} \mathbf{u} \quad \text{or} \quad \mathcal{Z} = \mathbf{S}^{-1} \mathcal{U}. \tag{17}$$

A matrix $\mathbf{G}$ consisting of eigenvalues $\omega_j$ along the diagonal (in increasing order) is formulated such that $\mathbf{G} = \mathbf{S}^{-1}\mathbf{A}\mathbf{S}$ which allows the transformation of Eq. (12) into the eigendomain as,

$$\frac{d^2 \mathbf{z}}{d\tau^2} = -\mathbf{G}\mathbf{z} + \mathbf{S}^{-1}\mathbf{M}^{-1}\mathbf{f} = -\mathbf{G}\mathbf{z} + \mathbf{h} \quad \text{or} \quad \frac{d^2 \mathcal{Z}}{d\tau^2} = -\mathbf{G}\mathcal{Z} + \mathbf{S}^{-1}\mathbf{M}^{-1}\mathcal{F} = -\mathbf{G}\mathcal{Z} + \mathcal{H} \tag{18}$$

The differential equations (18) are decoupled and can be solved to give,

$$\mathbf{z}(\tau) = \mathbf{C}^{(1)}\mathbf{a} + \mathbf{C}^{(2)}\mathbf{b} + \mathbf{z}_P(\tau) \quad \text{or} \quad \mathcal{Z}(\tau) = \mathbf{C}^{(1)}\mathcal{A} + \mathbf{C}^{(2)}\mathcal{B} + \mathcal{Z}_P(\tau) \tag{19}$$

where $\mathbf{C}^{(1)}$ is a diagonal matrix with $\mathbf{C}_{j,j}^{(1)} = \sin(\omega_j \tau)$, $\mathbf{C}^{(2)}$ is a diagonal matrix with $\mathbf{C}_{j,j}^{(2)} = \cos(\omega_j \tau)$, and $\mathbf{z}_P(\tau)$ or $\mathcal{Z}_P(\tau)$ are the particular solutions of the differential equations, which depend on $\mathbf{h}$ or $\mathcal{H}$ and, hence, depend on the external driving force $\mathbf{f}$ or $\mathcal{F}$. $\mathbf{a}$ or $\mathcal{A}$ and $\mathbf{b}$ or $\mathcal{B}$ are determined by the initial conditions from the initial displacement ($\mathbf{u}_o$ or $\mathcal{U}_o(k)$) and velocities ($\mathbf{v}_o$ or $\mathcal{V}_o(k)$).

$$\mathbf{b} = \mathbf{S}^{-1}\mathbf{u}_o - \mathbf{z}_P(0) \quad \text{or} \quad \mathcal{B} = \mathbf{S}^{-1}\mathbf{u}_o - \mathbf{z}_P(0) \tag{20}$$





and

$$\mathbf{a} = \mathbf{G}^{-1}\mathbf{S}^{-1}\mathbf{v}_o - \mathbf{G}^{-1}\frac{d\mathbf{z}_P(\tau)}{d\tau}\bigg|_{\tau=0} \quad \text{or} \quad \mathcal{A} = \mathbf{G}^{-1}\mathbf{S}^{-1}\mathcal{V}_o - \mathbf{G}^{-1}\frac{d\mathcal{Z}_P(\tau)}{d\tau}\bigg|_{\tau=0} \tag{21}$$

$\mathbf{a}$ and $\mathbf{b}$ or $\mathcal{A}$ and $\mathcal{B}$ are column vectors with column elements $a_j$ and $b_j$ or $\mathcal{A}_j$ and $\mathcal{B}_j$, associated with a particular eigenfrequency ($\omega_j$). The solution in real space can be obtained by the transformation mentioned in Eq. (17) which can be applied on Eq. (19) to give.

$$\mathbf{u}(\tau) = \mathbf{S}\mathbf{C}^{(1)}\mathbf{a} + \mathbf{S}\mathbf{C}^{(2)}\mathbf{b} + \mathbf{u}_P(\tau) \quad \text{or} \quad \mathcal{U}(\tau) = \mathbf{S}\mathbf{C}^{(1)}\mathcal{A} + \mathbf{S}\mathbf{C}^{(2)}\mathcal{B} + \mathcal{U}_P(\tau). \tag{22}$$

### 2.4 Initial Conditions : Impulse Driving

The initial conditions required to solve various special cases are the initial displacements ($\mathbf{u}_o$) and initial velocities ($\mathbf{v}_o$) in real space and $\mathcal{V}_o$ and $\mathcal{U}_o$ in spatial Fourier space. Besides the sinus driving used in Lawney and Luding (2014), we apply impulse driving initial condition. For an impulse driving mode, the boundary conditions are as follows,

$$u^{(0)}(\tau = 0) = 0, \ u^{(N+1)}(\tau = 0) = 0, \ v^{(1)}(\tau = 0) = v_o. \tag{23}$$

An impulse driven chain has an impulse imparted to the initial particle. The $i = 1$ particle has an initial velocity $v_o$. Using Eq. (22), (20), (21) and the initial conditions for the impulse driven chain i.e. $\mathbf{f} = 0$ (no driving present), $\mathbf{u}_o = 0$ and $\mathbf{v}_o = [v_o \ 0 \dots 0]^{\mathrm{T}}$, we get,

$$\mathbf{a} = \mathbf{G}^{-1}\mathbf{S}^{-1}\mathbf{v}_o \quad , \quad \mathbf{b} = 0, \tag{24}$$

$$\mathbf{u} = \mathbf{S}\mathbf{C}^{(1)}\mathbf{G}^{-1}\mathbf{S}^{-1}\mathbf{v}_o \quad \& \quad \mathbf{v} = \mathbf{S}\mathbf{C}^{(2)}\mathbf{S}^{-1}\mathbf{v}_o, \tag{25}$$

which implies that displacement and velocity of individual particles are,

$$u^{(p)}(\tau) = v_o \sum_{j=1}^{N} \frac{S_{pj}S_{1j}\sin(\omega_{(j)}\tau)}{\omega_{(j)}} \quad \& \quad v^{(p)}(\tau) = v_o \sum_{j=1}^{N} S_{pj}S_{1j}\cos(\omega_{(j)}\tau). \tag{26}$$

In wavenumber space (spatial Fourier transform), the initial condition is specified by $\mathcal{V}_o(k)$ which can be a sine or cosine function in terms of wavenumber ($k$). Using Eq. (22) and $\mathcal{V}_o(k)$ we get,

$$\mathcal{A} = \mathbf{G}^{-1}\mathbf{S}^{-1}\mathcal{V}_o(k) \quad , \quad \mathcal{B} = 0, \tag{27}$$

and thus

$$\mathcal{U} = \mathbf{S}\mathbf{C}^{(1)}\mathcal{A} \quad \& \quad \mathcal{V} = \mathbf{S}\mathbf{C}^{(2)}\mathbf{G}\mathcal{A}, \tag{28}$$





## 2.5 Mass Distribution, Disorder Parameter ($\xi$), Ensemble Averaging & Binning

The mass distribution of the monodisperse chain has been selected randomly from normal ($f^{(n)}(b)$), uniform ($f^{(u)}(b)$) and binary ($f^{(bi)}(b)$) distributions whose standard deviations give the measure of the disorder of mass in the chain ($\xi$). For instance, the normal distribution is given by,

$$f^{(n)}(b) = \frac{1}{\xi\sqrt{2\pi}} e^{\frac{(b-1)^2}{2\xi^2}}. \tag{29}$$

Higher disorder means that the difference between the lightest particle and the heaviest particle is very large. It was observed in Lawney and Luding (2013) that the three distributions showed quantitatively similar behavior if the first two moments of the distributions were the same. The details of the distributions are given in Appendix B. The first two moments of the aforementioned three distributions have been matched. Ensembles of chains with different realizations for a particular disorder and distribution have been taken into consideration. Angular brackets will be used to denote ensemble averaged physical quantities like $\langle \mathbf{u} \rangle$, $\langle E_{tot} \rangle$, etc. The first five scaled moments of the three distributions for different disorder (standard deviation) $\xi = 0$, $\xi = 0.1$, $\xi = 0.2$, $\xi = 0.35$, $\xi = 0.5$ and $\xi = 0.8$ are given in Table 1 (500 ensembles) and Table 3 (10000 ensembles).

## 2.6 Participation Ratio & Localization Length

The participation ratio ($P_j$) (introduced in Bell and Dean (1970) and used previously in Allen and Kelner (1998), Zeravcic et al. (2009)) is a crucial tool in determining the localization length ($\tilde{L}_j$) associated with a particular eigenmode. This localization length is the length beyond which elastic waves with a particular frequency become evanescent, i.e., they decay exponentially in a disordered system. It is instrumental in determining the length within which the elastic waves become confined in space and is dependent on the frequency and thus the eigenmode (Anderson (1958)). The participation ratio of eigenmode $j$ is defined as,

$$P_j = \frac{1}{\sum\limits_{i=1}^{N} (S_{ij})^4} \tag{30}$$

with the normalization condition on the eigenmodes $\sum\limits_{i=1}^{N} S_{ij}^2 = 1$. For one dimension the localization length is defined as $\tilde{L} = P_j \tilde{d}$ where $\tilde{d}$ is the particle center distance in equilibrium, i.e. under pre-compression. The localization length can now be non-dimensionalised by the internal particle scale of separation $\sim \tilde{d}$ to give $L_j \cong P_j$. As discussed and pointed out in Allen and Kelner (1998), the localization length of the lowest eigenmode is often attributed to the length of the chain (which would be regarded as a force chain in our analysis) and hence, it becomes important to find the localization length of an ordered chain,





$\xi = 0$ as reference. For an ordered chain $b^{(1)}, b^{(2)}, b^{(3)}, ..., b^{(N)} = 1$ and $\kappa = 1$, so,

$$\mathbf{A} = \begin{bmatrix} -2 & 1 & 0 & 0 & \cdots & 0 \\ 1 & -2 & 1 & 0 & \cdots & 0 \\ 0 & 1 & \ddots & 0 & \cdots & 0 \\ 0 & \cdots & 0 & \ddots & 0 & 1 \\ 0 & \cdots & 0 & 0 & 1 & -2 \end{bmatrix} \tag{31}$$

The eigenvalues of this matrix are $\omega_j^2 = 2\sin^2\left(\frac{j\pi}{N}\right)$ and its eigenvectors are

$\mathbf{s}_{(j)} = \{\sin\left(\frac{j\pi}{N}\right), \sin\left(\frac{2j\pi}{N}\right), \sin\left(\frac{3j\pi}{N}\right)...\sin\left(\frac{(N-1)j\pi}{N}\right)\}$. After respecting the normalization condition and the definition of the

participation factor, the localization length of the lowest eigenmode ($P_0$) can be analytically calculated from the eigenvectors

as,

$$P_{norm} = \sum_{i=1}^{i=N} \sin\left(\frac{ij\pi}{N}\right)^2, \text{ and hence, } P_0 = \frac{1}{\sum_{i=1}^{i=N} \left(\frac{\sin\left(\frac{ij\pi}{N}\right)}{\sqrt{P_{norm}}}\right)^4} \tag{32}$$

For $N = 256$, irrespective of $j = 1, 2, 3..N$, $P_0 = 170.667 \approx 171$.

## 3 Dispersion

The analytical expression for the dispersion relation in an ordered chain of particles/elements with linear contact is given by
(Brillouin (1946), Tournat et al. (2004), Lawney and Luding (2014)),

$$\tilde{\omega}^2 = 4\frac{\tilde{\kappa}_o}{\tilde{M}_1}\sin^2\left(\frac{\tilde{k}\tilde{d}}{2}\right), \tag{33}$$

where the wavenumber can be non-dimensionalized by the microscopic particle scale of separation ($\tilde{d}$) and frequency by $\sqrt{\frac{\tilde{\kappa}_o}{\tilde{M}_1}}$
giving the non-dimensional dispersion relation

$$\omega^2 = \Omega_\pi^2\sin^2\left(\frac{k}{2}\right). \tag{34}$$

with $\Omega_\pi = 2$ for ordered chains with $\xi = 0$. Eq. (34) holds for propagative as well as evanescent waves. The positive roots
of this relation correspond to propagative waves and the imaginary roots to evanescent waves (Tournat et al. (2004)). This
expression also holds for longitudinal wave propagation in 3D granular packings (Mouraille and Luding (2008)) and in 1D
chains as well (Lawney and Luding (2014)). From the dispersion relation, it can be noted that disorder creates a maximum
permissible frequency ($\Omega_\pi$) for propagating waves, frequencies below $\Omega_\pi$ are propagative until the order of their localization
length (Sect. 2.6) and the frequencies above $\Omega_\pi$ are evanescent. The dispersion relation (Eq. (34)) for ordered chains ($\xi = 0$) is

$$\omega = 2\sin\left(\frac{k}{2}\right), \tag{35}$$

which is the dispersion relation for propagative waves.





### 3.1 Total Energy Dispersion

From Eq. (A5) it can be observed that the total energy of the eigenmodes is constant with respect to time as given by,

$$E_{tot}(\omega_j, k) = \frac{1}{2}\mathcal{A}_j(k)^2\omega_j^2. \tag{36}$$

By taking the first moment of this eigenmodal total energy representation about frequency, a dominant frequency related to a
particular wavenumber can be obtained. Moments of the eigenmodal total energy representation are defined as,

$$M^{(m)}(k) = \frac{\sum \omega_j^m E_{tot}(\omega_j, k)}{E_{tot}(\omega_j, k)}. \tag{37}$$

The dominant frequency is given by the first moment,

$$\Omega(k) = M^{(1)}(k) = \frac{\frac{1}{2}\sum_j \mathcal{A}_j^2\omega_j^3}{E_{tot}}. \tag{38}$$

The dominant frequency can be measured by averaging over all eigenmodes for a single realization with, $\mathcal{A}_j(k)$ as a multi-
plicative factor which depends on the Fourier initial condition $\mathcal{V}_o(k)$ (Eq. (27)). The dispersion relation for the propagating
waves can be obtained by taking ensemble averages of this dominant frequency ($\langle\Omega(k)\rangle$), which will be plotted in Fig. 8(b)
below for different disorder strengths (500 ensembles).

### 3.2 Group velocity

The group velocity is given by,

$$v_g = \frac{\partial \omega}{\partial k}, \tag{39}$$

for both propagative waves and evanescent waves, it can be obtained by differentiating Eq. (35).

$$v_g(k) = \frac{\sqrt{\Omega_\pi^2 - \omega^2}}{2}. \tag{40}$$

where $\Omega_\pi(\xi)$ depends on disorder as we will see below.

### 4 Results & Discussions

The analytical expressions derived in the previous sections are computed for $N = 256$ particles long chains. The impulse
imparted to the first particle is, $v_o = 0.05$. The time step utilized for the output is, $\Delta t = 0.0312$ and the maximum time up to
which the computations have been carried out is, $t_{max} = 256$ such that the pulse has just about reached the 256$^{\text{th}}$ particle. As it
can be seen from Tables 1 and 3, the scaled average mass of the particles has been kept $M_1 = 1$ and $\xi = 0.1, 0.2, 0.35, 0.5$ and
0.8 disorder parameters (standard deviation; see appendix) have been used for analysis. 500 ensembles and 10000 ensembles
of chains along with representative single realizations will be shown in this section.



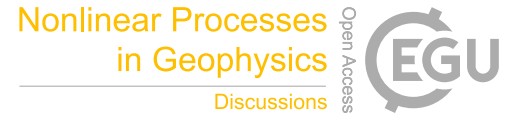

### 4.1 Space Time Response

Equation (26) has been used to generate the analytical space time responses of individual particles of the 256 particles long chain before ensemble averaging was carried out (500 and 10000 times).

#### 4.1.1 Displacement Response of the Three Distributions

Figure 3 shows the displacement in time of the $150^{th}$ particle (Fig. 1(a) & 1(c)) and of the $220^{th}$ particle (Fig. 1(b) & 1(d)), which are placed before and after the reference localization length (the maximum possible, $L_{max} = 171$, Sect. 4.4) for two disorder parameters, $\xi = 0.1$ and $\xi = 0.5$ with respect to the three mass distributions (normal, uniform and binary). For weak disorder ($\xi = 0.1$), it is observed that the displacement wave packets are perfectly superposed over each other affirming what was concluded in Lawney and Luding (2013) & Lawney and Luding (2014) that the shape of the distribution has no effect

on the propagating pulse if the first two moments of the distribution are the same (Table 1). For stronger disorder ($\xi = 0.5$), the wave packets are not collapsing perfectly over each other (Fig. 1(c) & 1(d)). As it can be seen in Table 2 that there is a numerical mismatch between the unscaled moments of the distributions leading to a dissimilarity between the second scaled moments ($\langle M_2 \rangle$). This also causes the real standard deviation (disorder) which has been numerically calculated ($\Xi$; Table 1) to deviate a little bit from it's intended value of $\xi$, however, the deviation is not a lot. It can also be observed from Fig. 3 (c)

and (d)) that the pulse shapes of normal distribution and uniform distribution are closer to each other in comparison to normal and binary or binary and uniform which stems from the fact that the scaled second moments ($\langle M_2 \rangle$) of normal and uniform distribution for $\xi = 0.5$ are closer to each other (Table 1) in comparison to the second scaled moment of the binary distribution. Similar conclusions about similarity, dissimilarity and closeness can be drawn about pulse shapes of different distributions for different disorder parameters ($\xi$ (intended), $\Xi$ (numerically obtained)) on the basis of moments of the mass distribution. For

larger $\xi$, higher moments have to be considered (Ogarko and Luding (2013)). Discussing the effect of higher moments goes beyond the scope of this study.



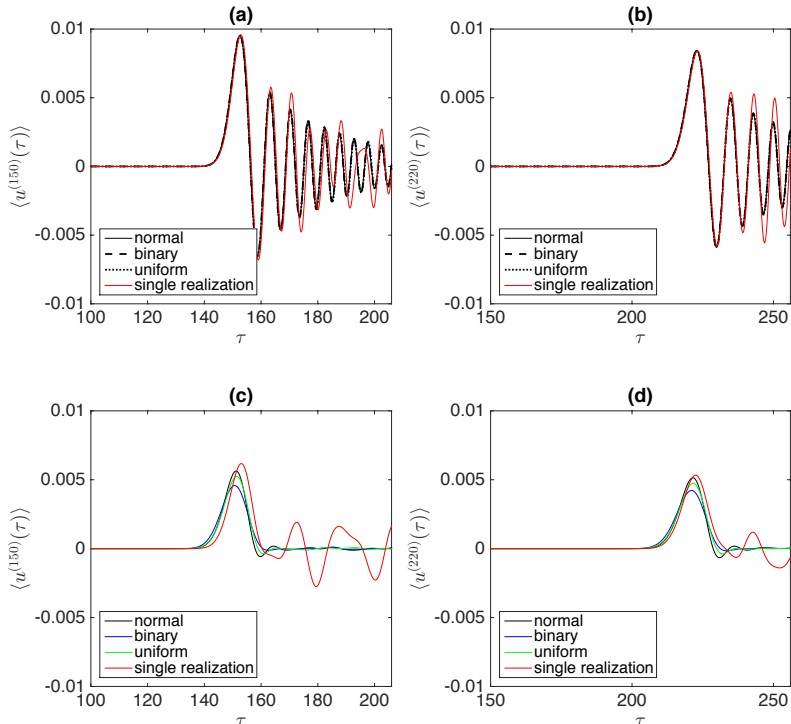

**Figure 3.** Ensemble averaged displacements (500 times) of $150^{th}$ (Fig. 3(a) and Fig. 3(c)) and $220^{th}$ (Figures 3(b) and 3(d)) particle with respect to time. Figures 3(a) and 3(b) have disorder parameter, $\xi = 0.1$, Fig. 3(c) and Fig. 3(d) have disorder parameter, $\xi = 0.5$.

### 4.1.2 Displacement Response for Different Disorder Parameters ($\xi$)

Mechanical waves propagating through disordered media or granular media like soil (on the receiver end) can be divided into two parts, the coherent part and the incoherent part (Jia et al. (1999), Jia (2004)). The coherent part is the leading wave packet and self averaging in nature (it maintains its shape after ensemble averaging), it is used for determining bulk sound wave veloc-

5  ity. In contrast, the incoherent part is the scattering, non-self averaging part, which is strongly system configuration dependent, also known as coda or tail of the mechanical wave. Figure 4 contains displacement of the same particles ($150^{th}$ & $220^{th}$ particles) used in subsection 4.1.1 for consistency. Here attention has been given to the effect of the mass distribution on the time of arrival/flight and hence, the wave velocity of the initial wave packet. Figures 4(a) and 4(b) contain the displacements of the $150^{th}$ and $220^{th}$ particle (before and after the $L_{max}$, Sect. 4.4) for single realization, 500 ensembles and 10000 ensembles. The

10  leading wave packet is the same for 500 ensembles and 10000 ensembles in both the figures, i.e. the coherent part of the wave which maintains its shape after averaging. The coda is more or less pronounced at 150 or 220 respectively and vanishes due to ensemble averaging. Figures 4(c) and 4(d) show the displacement response of $150^{th}$ and $220^{th}$ particles with respect to time



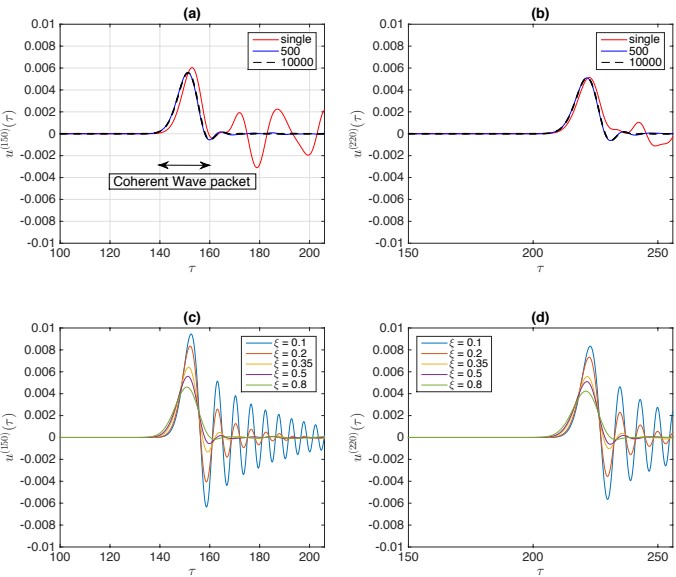

**Figure 4.** Displacements of 150[th] (Fig. 3(a) and Fig. 3(c)) and 220[th] (Fig. 3(b) and Fig. 3(d)) particle with respect to time for different ensembles (Fig. 3(a) and Fig. 3(b)) of disorder, $\xi = 0.5$ and of multiple disorder (Fig. 3(c) and Fig. 4(d); for 500 ensembles).

for chains with different mass disorder. The speed of the coherent wave packet (from source to receiving particle) is increasing with disorder. Higher disorder leads to higher coherent wave speed, irrespective of the localization length ($L_{max}$). However, this increase in wave speed can also be attributed to sound wave accelaration near the source as pointed out by Mouraille et al. (2006) and may not be generalized as effect of mass disorder in the chain, this has been investigated in the next section.

### 4.1.3 Coherent Wave Speed and Disorder

Table 4 contains the velocity of the peak of the coherent wave, the velocity of the rising part of the coherent wave packet when the displacement of the particle has attained 5%, 10%, 70%, 90% of the peak value and the first time when the displacement of the particle becomes 0 after it has attained the peak value of the coherent wave (zero crossing), all constituting the coherent wave packet. The results have also been plotted in Fig. 5. The velocities were determined through velocity picking (particle position divided by the time of arrival). The particles used for computing the velocities were 130, 150, 200 and 220 (2 before localization length and 2 after localization length, $L = 171$). It can be observed that irrespective of the rising part of the coherent wave packet (5%, 10%, of the peak value etc.) and the peak (Fig. 5(a), (b), (c), (d)), the wave velocity increases with disorder. However, for zero crossing (Fig. 5(e)), the velocity decreases with increase in disorder and the same can be said for the part of the coherent wave packet which lies after the peak value, this can be attributed to the increased spreading of the wave packet with increase in disorder. Fig. 5(f) shows the velocity of the peak value of the coherent wave packet of all the particles of



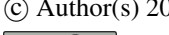

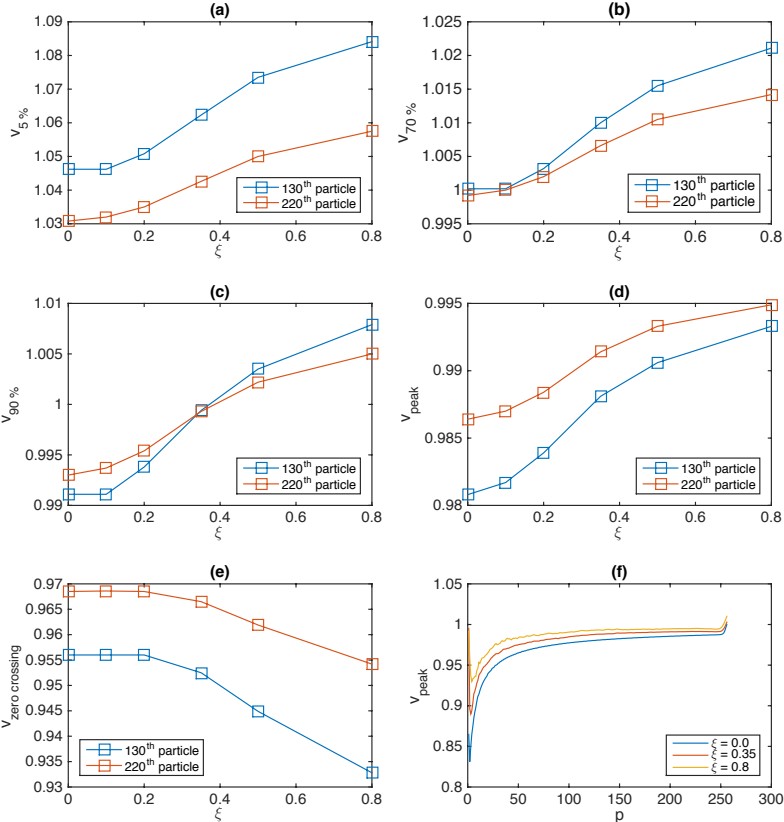

**Figure 5.** Coherent Wave Velocities determined through Velocity picking.

the granular chain for different disorder parameters and it also exhibits a similar kind of acceleration of signal/mechanical wave near the source as was observed in Mouraille et al. (2006). This acceleration is caused by self-demodulation of the initial impulse imparted to the granular chain and the noteworthy point is that the acceleration increases with increase in disorder. However, due to this observation we cannot generalize the effect of disorder on wave speed. The sudden rise in velocity of the peak value in Fig. 5(f) after the $250^{th}$ particle is due to the boundary effect as well as due to the presence of coherent wave front of the traveling wave around that position as the maximum time window used is $t_{max} = 256$.





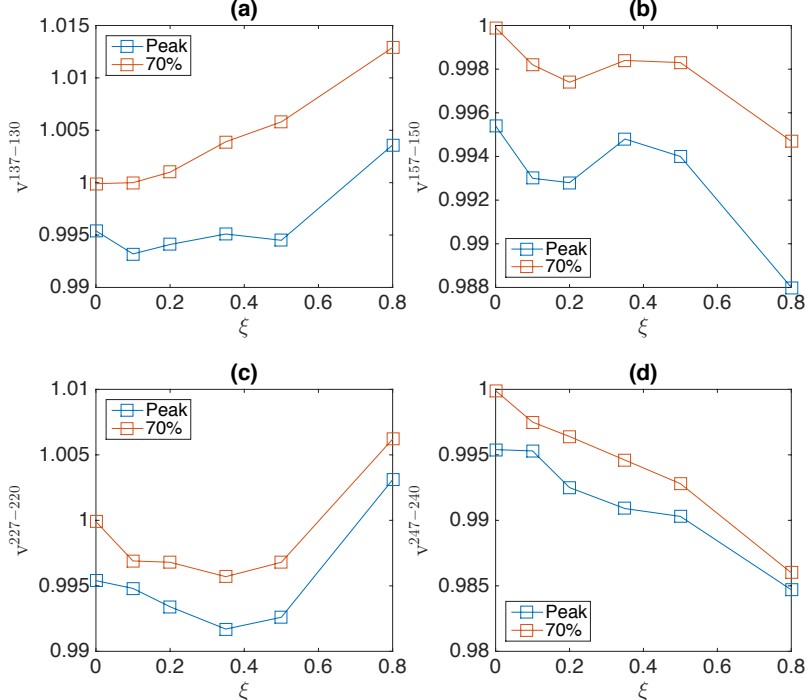

**Figure 6.** Wave speed for common distance of separation (7 particles/elements) with different disorder $\xi$.

To understand the effect of disorder on wave speed without taking into account this "source effect", the time taken by the pulse for propagating a common distance of separation has been computed and the velocity of the wave computed by using it (common distance of separation divided by the time taken by the pulse to travel) has been taken in Table 6 (7 particles as the common distance of separation), the results have also been plotted in Fig. 6. The reason for selecting such a low common

5   distance of separation was to keep the effect of the source as minimal as possible. The velocities have been computed from the time taken by the pulse to travel between different sets of points (130 to 137, 150 to 157, 220 to 227 and 240 to 247 particles) with use of different reference points of the coherent wave front (5%, 10%, 70%, 90%, peak value and zero crossing). From Fig. 6 and Table 6 it can be observed that the velocities computing using different reference points follow the same trend except the velocity computed using zero crossing reference point. Velocities computed using zero crossing reference point are more or

10  less constant with little fluctuations (Table 6). Fig. 6(a) shows a consistent increase of velocity as it is the closest to the source (dominated by the source effect), however, as the set of particles is selected farther away from the source, the velocity trend shows a slight decrease and then an increase with the increase in $\xi$ (Fig. 6(b) and (c)). Fig. 6(d) exhibits a consistent decrease of velocity with increase in $\xi$ because the set of particles (247-240) are far from the source (source effect is weak). From Fig. 6(d), it can be interpreted that higher disorder results in decrease in wave velocity.





## 4.2 Frequency Response & Dispersion

In Fig. 7(a) & Fig. 7(c) a Fast Fourier Transform (FFT) with respect to time has been carried out on the displacement response of a 256 element long chain for disorder, $\xi = 0.01$ and $\xi = 0.35$ respectively (when an impulse of $v_o = 0.05$ has been applied to the first particle) to observe the frequency content with distance, (the sampling frequency is $\omega_{sample} = \frac{2\pi}{\Delta t}$) and responses upto half of the sampling frequency were taken into account to avoid aliasing. The first 5 particles have been excluded from the Fourier transform to avoid the driving effect. Fig. 7(a) exhibits the existence of cut-off frequency ($\omega = 2$) above which the waves become evanescent. The bending of the density with distance (particle number) especially at large distances is attributed to dispersion and the finite time window which is well encapsulated (red curve) by the analytical solution given in Eq. (42) using the group velocity ($p$ is the particle number).

$$v_g t_{max} = p, \tag{41}$$

For an ordered chain ($\xi = 0$), $\Omega_\pi = 2$ and on using Eq (40), the previously mentioned equation can be written as,

$$\omega(p) = 2\sqrt{1 - \frac{p^2}{t_{max}^2}}, \tag{42}$$

which is the red curve plotted in Fig. 7(a).

A spatial as well as temporal 2D FFT has been carried out for a single realization of a 256 element long chain with disorder $\xi = 0$ and $\xi = 0.35$ to observe the dispersion relation (Fig. 7(b) & 7(d); $\omega$ v/s $k$). 2D FFT has been used previously for one-dimensional and three dimensional polydisperse granular packings for obtaining dispersion relations (Lawney and Luding (2014), O'Donovan et al. (2015), Luding and Mouraille (2008)) but strong frequency filtering due to the disordered system resulted in ambiguous dispersion relations (flat bands and absence of certain frequencies below the cut-off frequency which is the propagative band due to the presence of defect modes). This can also be observed from the density plots in the Fig. 7(b) and Fig. 7(d). Eq. (35) (the dispersion relation for an ordered chain) has been plotted in Fig. 7(b) which gives a perfect fit for the denser regime in the figure. However for the disordered chain, $\xi = 0.35$, as proposed earlier in Sect. 3.1, the dispersion relation can be given by $\langle \Omega(k) \rangle$ by ensemble averaging the dominant frequencies with respect to different wavenumbers. $\langle \Omega(k) \rangle$ for 500 ensembles with disorder $\xi = 0.35$ has been plotted in Fig. 7(b) (the green curve). It can be seen that for low frequencies the green curve perfectly superposes the dense regime in the density plot of displacement's temporal and spatial Fourier transform, for higher frequency due to the appearance of the flat band (defect mode) the density of frequencies are not present near the green curve. But, it can be noted that the green curve holds true for low and meso-level frequencies/wavenumbers.



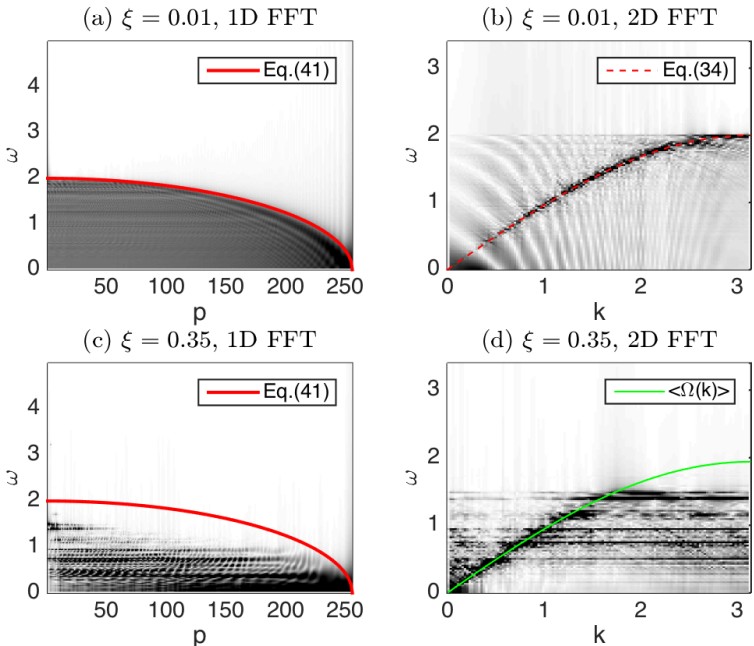

**Figure 7.** Figure 7(a) is the temporal Fourier transform of displacement of particles for disorder parameter $\xi = 0.01$ (single realization) with group velocity ($v_g$) depicting the propagation of wave front and Fig. 7(b) is the temporal as well as spatial Fourier transform (2 D FFT, single realization) calculated for obtaining dispersion relation of a chain, $\langle \Omega(k) \rangle$ gives the true dispersion relation. Fig. 7(c) and (d) are $\xi = 0.35$ counterparts of Fig. 7(a) and (b) respectively.

### 4.3 Total Energy Dispersion in Disordered Chains

The $\langle \Omega(k) \rangle$ which was plotted for $\xi = 0.35$ in Fig. 7(b) has been plotted for $\xi = 0.1, 0.2, 0.35, 0.5$ and $0.8$ in Fig. 8(a). It is observed that maximum permissible frequency ($\Omega_\pi$) above which the waves become evanescent decreases with increasing disorder. The slope of $\omega$ v/s $k$ curves indicates the wave speed, which clearly can be observed to be decreasing with increasing disorder. This indicates that the wave speed is decreasing with increasing disorder confirming what was observed in Sect. 4.1.3.

### 4.4 Participation Ratio & Localization length

Figure 9 shows the participation ratio ($\langle P \rangle$), i.e. the localization length ($\langle L \rangle$, from Sect. 2.6) for binned 500 ensemble averaged realizations of chains (with 0.0781 as frequency bin size) and with different disorder parameters, $\xi = 0.1, 0.2, 0.35, 0.5$ and $0.8$. The lowest frequencies have always the same localization length, which is $L_{max} = 171$, independent of the disorder of mass in the chain. In Sect. 2.6, $L_{max}$ has been derived analytically from an ordered chain and through the definition of the participation ratio. Towards higher frequency it can be observed that the localization length becomes same for all the disordered chains at a




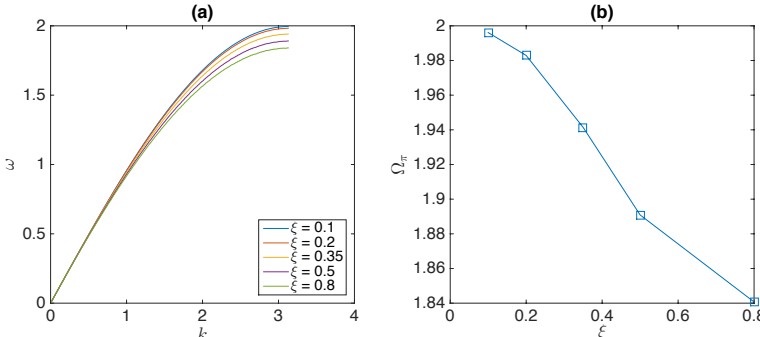

**Figure 8.** Dispersion relation (Fig. 8(a), $\langle\Omega(k)\rangle$) with respect to different wavenumbers and the maximum permissible frequency has been shown for disorder parameters, $\xi = 0.1, 0.2, 0.35, 0.5$ and $0.8$

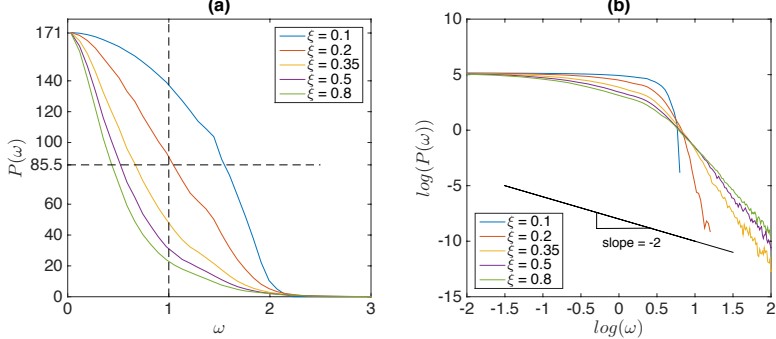

**Figure 9.** Participation Ratio or the Localization length with respect to different frequencies for 500 ensembles with $\xi = 0.1, 0.2, 0.35, 0.5$ and $0.8$ and bin size = 0.0781.

particular frequency (cross-over frequency). For frequencies above the cross-over frequency, the localization length increases with increasing disorder and for frequencies below the cross-over frequency, the localization length decreases with increasing disorder. Unlike infinitely long chains where $L \propto \omega^{-2}$ (as suggested in Azbel (1983)), the finite disordered chains for higher frequencies have $L \propto \omega^{-q}$ where $q > 2$ and decreases with increasing disorder.

5    For understanding the effect of disorder on cut-off frequency ($\omega_{pass}$) associated with the localization lengths and the localization length ($L$), $\omega_{pass(1/2)}$ (the cut-off frequency associated with half the value of $L_{max}$) for different disorder parameters have been plotted in Fig. 10(a) and $L$ associated with $\omega = 1$ scaled with $L_{max}$ for multiple disorder parameters have been plotted in Fig. 10(b). The values have been selected from the dashed lines in Fig. 9(a). Both of the quantities exhibit a decreasing trend with increase in disorder in the pass band of frequencies (0-$\Omega_\pi$).

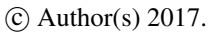



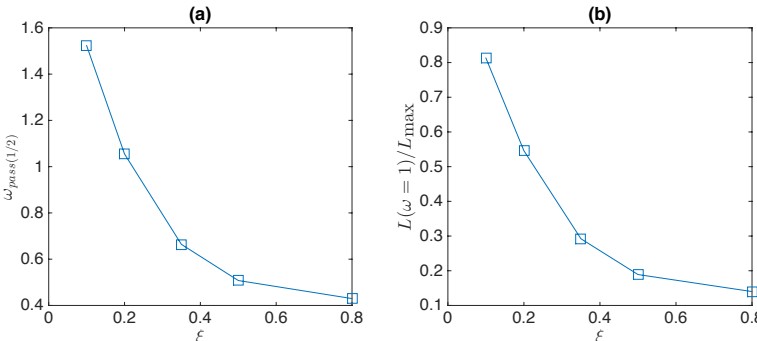

**Figure 10.** (a) $\omega_{pass(1/2)}$ for different $\xi$. (b) Localization length scaled with maximum localization length ($L_{max}$) for different $\xi$.

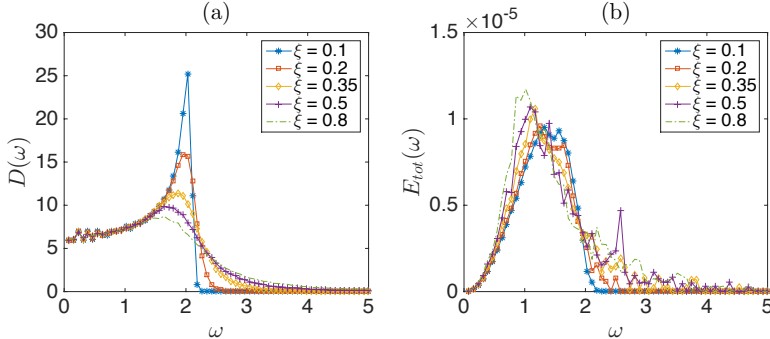

**Figure 11.** (a) Density of States (DOS) and (b) energies of the binned frequencies for different disorder $\xi$.

### 4.5 Total Energy of Eigenmodes

The Density Of States (DOS) or density of vibrational modes is treated as an important quantifying factor in studying the vibrational properties of materials like crystals, jammed granular media (Schreck et al. (2014)), etc. However, it tells us only about the number of vibrational modes but, does not paint the complete picture of spectral properties of vibration like energy
5  transport. Eq. (A5) gives us the total energy of individual eigenmodes and also from the equation, we can deduce that the energy is constant with respect to time. Figure 11(a) plots the ensemble averaged density of states for 500 mass disordered granular chains with frequency bins of size 0.0781. The peak of the density is decreasing with increasing disorder and shifting to smaller $\omega$. Figure 11(b) gives the ensemble averaged constant energy spectrum for the same frequency bins used in the previous plot (500 realizations) giving an energy distribution over frequency. The shape of the energy distribution is wider
10  over frequency for low $\xi$ but, for higher $\xi$, the energy distribution becomes more sharp with a peak. The peak increases with increasing disorder. In both plots Fig. 11 (a) and (b), the tails are broader for larger $\xi$.



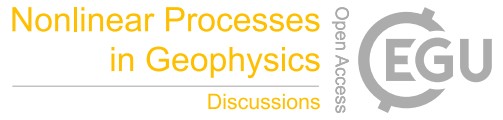

## 5 Conclusions

An impulse driven wave propagating through a precompressed mass disordered granular chain has been studied. Motivation comes from the existence of force chains, which form the backbone network for mechanical wave propagation in granular materials like soil. The scaled standard deviation of the mass probability distribution of the elements/particles of the granular chain

has been identified as the relevant disorder parameter ($\xi$; see Sect. 2.5), as suggested already in Lawney and Luding (2014). Chains with normal, binary and uniform mass distributions have quantitatively identical signal transmission characteristcs as long as the first two moments of the mass distribution are the same and $\xi$ is not too large.

Interestingly, on first sight, the dependence of wave speed on magnitude of disorder looks non-monotonous. This surprising increase of wave-speed for weak disorder, and decrease for stronger disorder, is due to two different effects overlapping: The

10 increase of wave-speed takes place close to the source, see Fig. 5, i.e. our 1D granular chain has the ability to model the physics of accelerating waves, as observed in higher dimensional, complex granular structures (Mouraille et al. (2006)). The competing mechanism of decreasing wave-speed with disorder is only clearly observed when the velocities are measured as travel time with maintaining constant distance of separation far away from the source (Fig 6 and Table 6). When the travel time is measured from the source, the two mechanisms overlap and interfere, causing the non-monotonous behavior, but possibly

allowing for tuning particular propagation characteristics in short chains.

As another main result, Eq. (38) gives an effective, weighted dispersion relation as the normalized first moment of eigen-modal (total) energies with frequency. This gives a much better signal to noise ratio for $\omega$ v/s $k$ in comparison to 2D FFT of displacement or velocity signals, reported previously ((Mouraille et al., 2006), O'Donovan et al. (2015)). Figure 8 shows that the upper limit (maximum permissible) frequency due to the discreteness of the system slightly decreases with increasing dis-

20 order, $\xi$, and consistently, waves propagate slightly slower with increasing disorder. From the energy content one also observes (in disordered systems) that waves above a low frequency pass-band ($\omega_{pass}$) become evanescent after they have traversed a localization length, $L = L(\omega)$, associated with this particular cut-off frequency.

The energy analysis used in this article can be used for understanding pulse propagation in disordered, weakly or strongly non-linear granular chains and its attenuation, widening and acceleration (experimentally and numerically investigated in Lan-

25 glois and Jia (2015)). An interesting question would be the effect of damping on the eigenmodes, velocity of the propagating wave, change in frequency filtering and the energy of the eigenmodes. Also, a different kind of averaging (micro-macro transition) should be developed using frequency bands to develop a Master Equation for propagation (or localization) of total energy in terms of wavenumber and frequency at different regimes of disorder, non-linearity, and materials. Such models, taking into account multiple scattering, dispersion, attenuation, etc. will allow for large system models of realistic wave propagation in

granular materials like soil.

## 6 Data availability

Data has been generated using the aforementioned theoretical model. The readers can reproduce it by using the equations mentioned in their respective sections.



## Appendix A: Total Energy Harmonic Evolution

The Energy of the system (chain) can be calculated by vector multiplications at a particular instance of time, the non-unitary dimension of the vector gives the respective information of the individual particles. The Kinetic Energy of the chain at a particular instant of time is,

$$E_{kin}(\tau) = \frac{1}{2}\mathbf{v}^{\mathrm{T}}\mathbf{Mv} \tag{A1}$$

Starting from the impulse initial condition in Sect. 2.4 and using $\mathbf{v} = \mathbf{SC}^{(2)}\mathbf{Ga}$ (Eq. (24) and (25)) and the orthonormality condition $\mathbf{S}^{\mathrm{T}}\mathbf{MS} = \mathbf{I}$ (Eq. (16)), where $\mathbf{I}$ is the identity matrix, the above equation becomes

$$
\begin{aligned}
E_{kin}(\tau) &= \frac{1}{2}(\mathbf{SC}^{(2)}\mathbf{Ga})^{\mathrm{T}}\mathbf{M}(\mathbf{SC}^{(2)}\mathbf{Ga}) \\
&= \frac{1}{2}\mathbf{a}^{\mathrm{T}}\mathbf{G}^{\mathrm{T}}(\mathbf{C}^{(2)})^{\mathrm{T}}\mathbf{S}^{\mathrm{T}}\mathbf{MSC}^{(2)}\mathbf{Ga} = \frac{1}{2}\mathbf{a}^{\mathrm{T}}\mathbf{G}\{\mathbf{C}^{(2)}\}^2\mathbf{Ga} = \frac{1}{2}\sum_j a_j^2\omega_j^2\sin^2(\omega_j\tau)
\end{aligned}
\tag{A2}
$$

Since $\mathbf{C}^{(1)}$, $\mathbf{C}^{(2)}$ and $\mathbf{G}$ are diagonal matrices, hence their transposition are equal to their original matrices. Note that there is no summation convention applied here. The Potential Energy of the chain at a particular instant of time is,

$$E_{pot}(\tau) = -\frac{1}{2}\mathbf{u}^{\mathrm{T}}\mathbf{Ku}, \tag{A3}$$

Using $\mathbf{u} = \mathbf{SC}^{(1)}\mathbf{a}$, $\mathbf{v} = \mathbf{SC}^{(2)}\mathbf{Ga}$, Eq. (25), and orthonormality, the above equation can be written as,

$$
\begin{aligned}
E_{pot}(\tau) &= -\frac{1}{2}\mathbf{u}^{\mathrm{T}}\mathbf{Ku} \\
&= -\frac{1}{2}\mathbf{u}^{\mathrm{T}}\mathbf{M}\frac{\mathrm{d}^2\mathbf{u}}{\mathrm{d}\tau^2} \\
&= -\frac{1}{2}(\mathbf{SC}^{(1)}\mathbf{a})^{\mathrm{T}}\mathbf{M}\frac{\mathrm{d}\mathbf{v}}{\mathrm{d}\tau} \\
&= -\frac{1}{2}(\mathbf{SC}^{(1)}\mathbf{a})^{\mathrm{T}}\mathbf{M}\frac{\mathrm{d}\mathbf{SC}^{(2)}\mathbf{Ga}}{\mathrm{d}\tau} \\
&= \frac{1}{2}\mathbf{a}^{\mathrm{T}}\mathbf{C}^{(1)}\mathbf{S}^{\mathrm{T}}\mathbf{MSC}^{(1)}\{\mathbf{G}\}^2\mathbf{a} \\
&= \frac{1}{2}\mathbf{a}^{\mathrm{T}}\mathbf{G}\{\mathbf{C}^{(1)}\}^2\mathbf{Ga} = \frac{1}{2}\sum_j a_j^2\omega_j^2\cos^2(\omega_j\tau).
\end{aligned}
\tag{A4}
$$

Hence, the Total Energy becomes a sum over all eigenmode energies,

$$E_{tot}(\tau) = \frac{1}{2}\sum_j a_j^2\omega_j^2. \tag{A5}$$

We can see that $E_{tot}$ is independent of time (the energy of our chain is conserved). This equation (A5) also gives us energy with respect to different eigenmodes of the chain (if we drop the first summation term). Hence, $E_{tot}(\omega_j) = \frac{1}{2}a_j^2\omega_j^2$. Now, by replacing $\mathbf{u}, \mathbf{v}, \mathbf{a}$ with their spatial Fourier transform counter parts $\mathcal{U}, \mathcal{V}$ and $\mathcal{A}$ (calligraphic) by using ansatz in spatial Fourier space as in Eq. (14) for Eq. (13), we obtain the harmonic total energy (in terms of wavenumber),

$$E_{tot}(\omega_j, k) = \frac{1}{2}\mathcal{A}_j^2(k)\omega_j^2. \tag{A6}$$





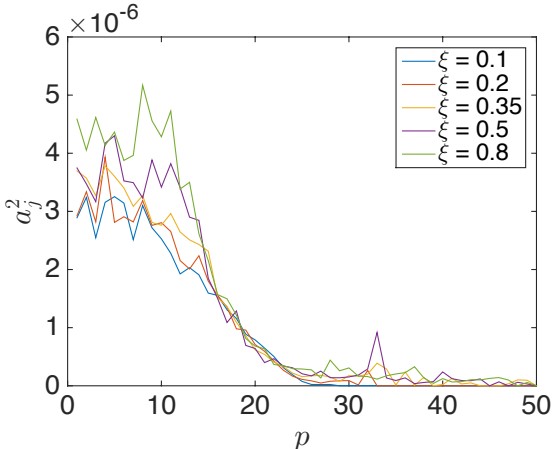

**Figure A1.** Multiplicative Factor $a_j$

**Appendix B: Matching the first two moments of different distributions (normal, uniform and binary).**

The raw $n^{th}$ moment of a distribution is defined as :

$$\tilde{M}_n^{(q)} = \int\limits_{-\infty}^{\infty} \tilde{m}^n f^{(q)}(\tilde{m}) d\tilde{m}, \tag{B1}$$

where, $f^{(q)}(\tilde{m})$ is the distribution, $q$ is the type of distribution and $\tilde{m}$ is the variable for which the distribution has been defined.

5   $q$ is $n$ for normal distribution, $u$ for uniform distribution and $bi$ for binary distribution. The raw scaled moment is defined as,

$$
\begin{aligned}
M_n^{(q)} &= \frac{\tilde{M}_n^{(q)}}{\tilde{M}_1^{(q)}}, \\
&= \frac{\int_{-\infty}^{\infty} \tilde{m}^n f(\tilde{m}) d\tilde{m}}{(\int_{-\infty}^{\infty} \tilde{m} f(\tilde{m}) d\tilde{m})^n}, \\
&= \frac{\int_{-\infty}^{\infty} \tilde{m}^n f(\tilde{m}) d\tilde{m}}{(\tilde{M}_1)^n}, \text{ (where first raw moment is the average of the distribution}(\tilde{M}_1)) \\
&= \int\limits_{-\infty}^{\infty} \left(\frac{\tilde{m}}{\tilde{M}_1}\right)^n f(\tilde{m}) d\tilde{m}, \\
&= \int\limits_{-\infty}^{\infty} b^n \{\tilde{m}_o f(\tilde{m})\} db, \text{ (where } b = \tilde{m}/\tilde{m}_o \text{ is the scaled mass (Sect. 2.1))} \\
&= \int\limits_{-\infty}^{\infty} b^n f(b) db. \text{ (}f(b) \text{ is the scaled mass distribution)}
\end{aligned}
\tag{B2}
$$

In case of particle mass distribution, only positive values can be considered so that the lower limit is to be replaced by zero, which leads to modifications for larger $\xi$.

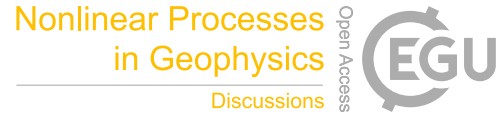
### B1 Normal Distribution

The unscaled normal distribution is given as :

$$f^{(n)}(\tilde{m}) = \frac{1}{\tilde{\xi}^{(n)}\sqrt{2\pi}} e^{-\frac{(\tilde{m}-\tilde{M}_1)^2}{2(\tilde{\xi}^{(n)})^2}}, \tag{B3}$$

where, $\tilde{\xi}^{(n)}$ is the standard deviation and $\tilde{M}_1$ is the average of the distribution. The scaled normal distribution is given as,

$$f^{(n)}(b) = \tilde{M}_1 f^{(n)}(\tilde{m}),$$

$$\phantom{f^{(n)}(b)} = \frac{1}{\xi^{(n)}\sqrt{2\pi}} e^{-\frac{(b-1)^2}{2(\xi^{(n)})^2}}. \tag{B4}$$

where, $b = \tilde{m}/\tilde{M}_1$ is the scaled mass and $\xi^{(n)} = \tilde{\xi}^{(n)}/\tilde{M}_1$ is the scaled standard deviation which is the disorder parameter for the one dimensional chain.

### B1.1 First Moment

The first scaled moment of the normal distribution is given as :

$$M_1^{(n)} = \int_{-\infty}^{\infty} b \frac{1}{\xi^{(n)}\sqrt{2\pi}} e^{-\frac{(b-1)^2}{2(\xi^{(n)})^2}} db,$$

$$= \underbrace{\int_{-\infty}^{\infty} (b-1) \frac{1}{\xi^{(n)}\sqrt{2\pi}} e^{-\frac{(b-1)^2}{2(\xi^{(n)})^2}} db}_{\text{non-even power of b}} + \underbrace{\int_{-\infty}^{\infty} \frac{1}{\xi^{(n)}\sqrt{2\pi}} e^{-\frac{(b-1)^2}{2(\xi^{(n)})^2}} db}_{\text{even power of b}},$$

$$= 0 + \frac{1}{\sqrt{\pi}} \times \sqrt{\pi}$$

$$= 1. \tag{B5}$$

Hence, the first scaled moment of the normal distribution is 1.

### B1.2 Second Moment

The Gaussian integral (normalizing condition) is given by :

$$\frac{1}{\sqrt{2\pi(\xi^{(n)})^2}} \int_{-\infty}^{\infty} e^{-\frac{(b-1)^2}{2(\xi^{(n)})^2}} db = 1, \tag{B6}$$

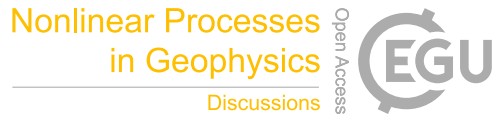



it can be differentiated with respect to $(\xi^{(n)})^2$ to get,

$$-\frac{1}{2(\xi^{(n)})^3\sqrt{2\pi}}\int\limits_{-\infty}^{\infty}e^{-\frac{(b-1)^2}{2(\xi^{(n)})^2}}\,db+\frac{1}{\sqrt{2\pi(\xi^{(n)})^2}}\int\limits_{-\infty}^{\infty}\frac{(b-1)^2}{2(\xi^{(n)})^4}e^{-\frac{(b-1)^2}{2(\xi^{(n)})^2}}\,db=0,$$

$$\Rightarrow\frac{1}{\sqrt{(\xi^{(n)})^2 2\pi}}\int\limits_{-\infty}^{\infty}(b-1)^2 e^{-\frac{(b-1)^2}{2(\xi^{(n)})^2}}\,db=\frac{(\xi^{(n)})^2}{\sqrt{2\pi(\xi^{(n)})^2}}\underbrace{\int\limits_{-\infty}^{\infty}e^{-\frac{(b-1)^2}{2(\xi^{(n)})^2}}\,db}_{\text{Normalizing condition = 1}},$$

$$\Rightarrow\frac{1}{\sqrt{(\xi^{(n)})^2 2\pi}}\int\limits_{-\infty}^{\infty}b^2 e^{-\frac{(b-1)^2}{2(\xi^{(n)})^2}}\,db=(\xi^{(n)})^2-\underbrace{\frac{1}{\sqrt{2\pi(\xi^{(n)})^2}}\int\limits_{-\infty}^{\infty}e^{-\frac{(b-1)^2}{2(\xi^{(n)})^2}}\,db}_{\text{Normalizing condition = 1}}+\frac{1}{\sqrt{2\pi(\xi^{(n)})^2}}\int\limits_{-\infty}^{\infty}2b\,e^{-\frac{(b-1)^2}{2(\xi^{(n)})^2}}\,db$$

$$\Rightarrow M_2^{(n)}=(\xi^{(n)})^2-1+2\underbrace{\frac{1}{\sqrt{2\pi(\xi^{(n)})^2}}\int\limits_{-\infty}^{\infty}b\,e^{-\frac{(b-1)^2}{2(\xi^{(n)})^2}}\,db}_{M_1^{(n)}=1},$$

$$\Rightarrow M_2^{(n)}=1+(\xi^{(n)})^2. \tag{B7}$$

Taking $\xi^{(n)}=\xi$, the second scaled moment of the normal distribution is $1+\xi^2$.

### B2 Binary Distribution

5   The unscaled binary distribution is given by :

$$f^{(bi)}(\tilde{m})=\frac{\delta(\tilde{m}-(\tilde{M}_1+\tilde{\xi}^{(bi)}))}{2}+\frac{\delta(\tilde{m}-(\tilde{M}_1-\tilde{\xi}^{(bi)}))}{2}, \tag{B8}$$

The scaled binary distribution is given as :

$$f^{(bi)}(b)=\tilde{M}_1 f^{(bi)}(\tilde{m}),$$
$$=\frac{\tilde{M}_1}{2}\left[\delta\left(\tilde{M}_1\left\{\frac{\tilde{m}}{\tilde{M}_1}-\left(\frac{\tilde{m}}{\tilde{M}_1}-\frac{\tilde{\xi}^{(bi)}}{\tilde{M}_1}\right)\right\}\right)+\delta\left(\tilde{M}_1\left\{\frac{\tilde{m}}{\tilde{M}_1}-\left(\frac{\tilde{m}}{\tilde{M}_1}-\frac{\tilde{\xi}^{(bi)}}{\tilde{M}_1}\right)\right\}\right)\right],$$
$$=\frac{\delta\{b-(1-\xi^{(bi)})\}+\delta\{b-(1+\xi^{(bi)})\}}{2} \tag{B9}$$

where, $b=\tilde{m}/\tilde{M}_1$ is the scaled mass and $\xi^{(bi)}=\tilde{\xi}^{(bi)}/\tilde{M}_1$ is the scaled standard deviation which is the disorder parameter for

10   the one dimensional chain.



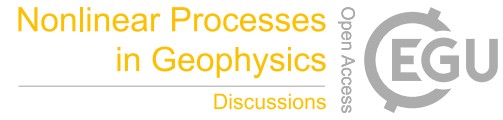

### B2.1 First Moment

The first scaled moment of the distribution is given as :

$$
\begin{aligned}
M_1^{(bi)} &= \int_{-\infty}^{\infty} b f^{(bi)}(b) db, \\
&= \int_{-\infty}^{1} b f^{(bi)}(b) db + \int_{1}^{\infty} b f^{(bi)} db, \text{ (Splitting the integral around 1)} \\
&= \int_{-\infty}^{1} b \frac{\delta\{b - (1 - \xi^{(bi)})\}}{2} db + \int_{1}^{\infty} b \frac{\delta\{b - (1 + \xi^{(bi)})\}}{2} db, \text{ (Probabilty distribution function)} \\
&= \frac{1 - \xi^{(bi)}}{2} + \frac{1 + \xi^{(bi)}}{2}, \\
&= 1.
\end{aligned}
\tag{B10}
$$

Hence, the first scaled moment of the binary distribution is 1.

### 5  B2.2 Second Moment

The second scaled moment of the binary distribution is given as follows :

$$
\begin{aligned}
M_2^{(bi)} &= \int_{-\infty}^{\infty} b^2 f^{(bi)}(b) db, \\
&= \int_{-\infty}^{1} b f^{(bi)}(b) db + \int_{1}^{\infty} b f^{(bi)} db, \text{ (Splitting the integral around 1)} \\
&= \int_{-\infty}^{1} b^2 \frac{\delta\{b - (1 - \xi^{(bi)})\}}{2} db + \int_{1}^{\infty} b^2 \frac{\delta\{b - (1 + \xi^{(bi)})\}}{2} db, \text{ (Probabilty distribution function)} \\
&= \frac{(1 - \xi^{(bi)})^2}{2} + \frac{(1 + \xi^{(bi)})^2}{2}, \\
&= 1 + (\xi^{(bi)})^2.
\end{aligned}
\tag{B11}
$$

Taking $\xi^{(n)} = \xi^{(bi)} = \xi$, the second scaled moment of the binary distribution is $1 + \xi^2$.

### B3  Uniform Distribution

10  The unscaled uniform distribution for the mass distribution is given by :

$$
f^{(u)}(\tilde{m}) = \begin{cases} \frac{1}{2\tilde{\xi}^{(u)}} & \text{for } \tilde{M}_1 - \tilde{\xi}^{(u)} \leq \tilde{m} \leq \tilde{M}_1 + \tilde{\xi}^{(u)} \\ 0 & \text{for } \tilde{m} < \tilde{M}_1 - \tilde{\xi}^{(u)} \text{ or } \tilde{m} > \tilde{M}_1 + \tilde{\xi}^{(u)} \end{cases}
\tag{B12}
$$

The value of the mass is $\frac{1}{2\tilde{\xi}^{(u)}}$ in the interval $\left[ \tilde{M}_1 - \tilde{\xi}^{(u)}, \tilde{M}_1 + \tilde{\xi}^{(u)} \right]$ and 0 elsewhere. The scaled uniform distribution is given as :

$$f^{(u)}(b) = \tilde{M}_1 f^{(u)}(\tilde{m}),$$

$$= \begin{cases} \frac{1}{2\xi^{(u)}} & \text{for } 1 - \xi^{(u)} \leq b \leq 1 + \xi^{(u)} \\ 0 & \text{for } b < 1 - \xi^{(u)} \text{ or } b > 1 + \xi^{(u)} \end{cases} \tag{B13}$$

The scaled masses ($b$) are selected from $1 \pm \xi$ to approximately p-reserve symmetry about the scaled mean.

5   **B3.1   First Moment**

The first scaled moment of the distribution is given as :

$$M_1^{(u)} = \int\limits_{-\infty}^{\infty} b f^{(u)}(b) db$$

$$= \int\limits_{-\infty}^{1-\xi^{(u)}} b f^{(u)}(b) db + \int\limits_{1-\xi^{(u)}}^{1+\xi^{(u)}} b f^{(u)}(b) db + \int\limits_{1+\xi^{(u)}}^{\infty} b f^{(u)}(b) db$$

$$= 0 + \int\limits_{1-\xi^{(u)}}^{1+\xi^{(u)}} b f^{(u)}(b) db + 0 \, (\text{The distribution is 0 elsewhere except the interval (B13)})$$

$$= \int\limits_{1-\xi^{(u)}}^{1+\xi^{(u)}} \frac{b}{2\xi^{(u)}} db$$

$$= \frac{b^2}{4\xi^{(u)}} \Bigg|_{1-\xi^{(u)}}^{1+\xi^{(u)}}$$

$$= 1. \tag{B14}$$

Hence, the first scaled moment of the uniform distribution is 1.



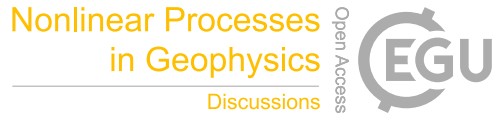

## B3.2   Second Moment

The second moment of the distribution is given as :

$$
\begin{aligned}
M_2^{(u)} &= \int\limits_{-\infty}^{\infty} b^2 f^{(u)}(b)\,db \\
&= \int\limits_{-\infty}^{1-\xi^{(u)}} b^2 f^{(u)}\,db + \int\limits_{1-\xi^{(u)}}^{1+\xi^{(u)}} b^2 f^{(u)}\,db + \int\limits_{1+\xi^{(u)}}^{\infty} b^2 f^{(u)}\,db \\
&= 0 + \int\limits_{1-\xi^{(u)}}^{1+\xi^{(u)}} b^2 f^{(u)}\,db + 0 \ \text{(The distribution is 0 elsewhere except the interval (B13))} \\
&= \int\limits_{1-\xi^{(u)}}^{1+\xi^{(u)}} \frac{b^2}{2\xi}\,db \\
&= \frac{b^3}{6\xi^{(u)}}\Bigg|_{1-\xi^{(u)}}^{1+\xi^{(u)}} \\
&= 1 + \frac{(\xi^{(u)})^2}{3}
\end{aligned}
\tag{B15}
$$

Taking $\xi^{(u)} = \sqrt{3}\xi^{(n)} = \sqrt{3}\xi^{(bi)} = \sqrt{3}\xi$

$$
\begin{aligned}
M_2^{(u)} &= 1 + \frac{(\xi^{(u)})^2}{3} \ \text{(from equation (B15))} \\
&= 1 + \frac{(\sqrt{3}\xi^{(n)})^2}{3} \ \text{(Replacement)} \\
&= 1 + \xi^2,
\end{aligned}
\tag{B16}
$$

thereby, placing a limit on the uniform distribution ($[1-\sqrt{3}\xi, 1+\sqrt{3}\xi]$) so that the intention of keeping the first two moments of three distribution is preserved.

From equations (B4), (B10) and (B14), it can be said that the first moment of the distributions have been matched. From equations (B7), (B11) and after a placing a limit on the uniform distribution, equation(B16) shows that the second moments of the distributions are matched. However, for large disorder, there is a need for correction as $b > 0$ and cannot be negative.

*Author contributions.*   The manuscript was prepared by R. Shrivastava and was co-authored by S. Luding.



*Acknowledgements.* This work is part of the Industrial Partnership Programme (IPP) 'Computational sciences for energy research' of the Foundation for Fundamental Research on Matter (FOM), which is part of the Netherlands Organisation for Scientific Research (NWO). This research programme is co-financed by Shell Global Solutions International B.V.

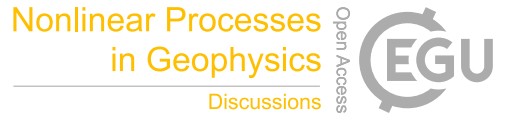

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



**Table 1.** Scaled Moments of ensemble averaged distributions (500 ensembles) used for the one-dimensional chain (256 element long).

| Distribution | Disorder | $< M_1 >$ | $< M_2 >$ | $< M_3 >$ | $< M_4 >$ | $< M_5 >$ | $\Xi$ | $\Xi^2$ |
|---|---|---|---|---|---|---|---|---|
| | $\xi = 0.0$ | 1.0000 | 1.0000 | 1.0000 | 1.0000 | 1.0000 | 0.0000 | 0.0000 |
| | $\xi = 0.1$ | 1.0000 | 1.0099 | 1.0298 | 1.0600 | 1.1010 | 0.1 | 0.0100 |
| Normal | $\xi = 0.2$ | 1.0000 | 1.0398 | 1.1194 | 1.2436 | 1.4219 | 0.1999 | 0.0400 |
| Distribution | $\xi = 0.35$ | 1.0000 | 1.1190 | 1.3590 | 1.7630 | 2.4184 | 0.3462 | 0.1195 |
| | $\xi = 0.5$ | 1.0000 | 1.2053 | 1.6366 | 2.4335 | 3.8973 | 0.4661 | 0.2061 |
| | $\xi = 0.8$ | 1.0000 | 1.3055 | 2.0104 | 3.5037 | 6.7333 | 0.6415 | 0.3067 |
| | $\xi = 0.0$ | 1.0000 | 1.0000 | 1.0000 | 1.0000 | 1.0000 | 0.0000 | 0.0000 |
| | $\xi = 0.1$ | 1.0000 | 1.0092 | 1.0283 | 1.0582 | 1.0999 | 0.0924 | 0.0092 |
| Binary | $\xi = 0.2$ | 1.0000 | 1.0399 | 1.1262 | 1.2682 | 1.4786 | 0.1849 | 0.0400 |
| Distribution | $\xi = 0.35$ | 1.0000 | 1.1388 | 1.4592 | 2.0373 | 2.9980 | 0.3235 | 0.1393 |
| | $\xi = 0.5$ | 1.0000 | 1.3250 | 2.1298 | 3.7514 | 6.8483 | 0.4621 | 0.3263 |
| | $\xi = 0.8$ | 1.0000 | 2.1359 | 5.4389 | 14.2177 | 37.4284 | 0.7394 | 1.1404 |
| | $\xi = 0.0$ | 1.0000 | 1.0000 | 1.0000 | 1.0000 | 1.0000 | 0.0000 | 0.0000 |
| | $\xi = 0.1$ | 1.0000 | 1.0100 | 1.0300 | 1.0602 | 1.1009 | 0.1002 | 0.0100 |
| Uniform | $\xi = 0.2$ | 1.0000 | 1.0400 | 1.1201 | 1.2431 | 1.4148 | 0.2004 | 0.0402 |
| Distribution | $\xi = 0.35$ | 1.0000 | 1.1227 | 1.3682 | 1.7639 | 2.3646 | 0.3508 | 0.1232 |
| | $\xi = 0.5$ | 1.0000 | 1.2508 | 1.7529 | 2.6212 | 4.0859 | 0.5011 | 0.2517 |
| | $\xi = 0.8$ | $---$ | $---$ | $---$ | $---$ | $---$ | $---$ | $---$ |





**Table 2.** Unscaled Moments of ensemble averaged distributions (500 ensembles) used for the one-dimensional chain (256 element long).

| Distribution | Disorder | $< \tilde{M}_1 >$ | $< \tilde{M}_2 >$ | $< \tilde{M}_3 >$ | $< \tilde{M}_4 >$ | $< \tilde{M}_5 >$ | $\tilde{\Xi}$ | $\tilde{\Xi}^2$ |
|---|---|---|---|---|---|---|---|---|
| | $\xi = 0.0$ | 1.0000 | 1.0000 | 1.0000 | 1.0000 | 1.0000 | 0.0000 | 0.0000 |
| | $\xi = 0.1$ | 1.0000 | 1.0100 | 1.0299 | 1.0601 | 1.1013 | 0.0999 | 0.0100 |
| Normal | $\xi = 0.2$ | 1.0000 | 1.0399 | 1.1197 | 1.2443 | 1.4232 | 0.1999 | 0.0400 |
| Distribution | $\xi = 0.35$ | 1.0022 | 1.1242 | 1.3689 | 1.7807 | 2.4492 | 0.3462 | 0.1195 |
| | $\xi = 0.5$ | 1.0274 | 1.2728 | 1.7768 | 2.7163 | 4.4725 | 0.4661 | 0.2061 |
| | $\xi = 0.8$ | 1.1581 | 1.7540 | 3.1363 | 6.3458 | 14.1470 | 0.6415 | 0.3067 |
| | $\xi = 0.0$ | 1.0000 | 1.0000 | 1.0000 | 1.0000 | 1.0000 | 0.0000 | 0.0000 |
| | $\xi = 0.1$ | 0.9618 | 0.9337 | 0.9151 | 0.9059 | 0.9058 | 0.0924 | 0.0092 |
| Binary | $\xi = 0.2$ | 0.9236 | 0.8873 | 0.8879 | 0.9240 | 0.9956 | 0.1849 | 0.0400 |
| Distribution | $\xi = 0.35$ | 0.8644 | 0.8553 | 0.9503 | 1.1501 | 1.4663 | 0.3235 | 0.1393 |
| | $\xi = 0.5$ | 0.8091 | 0.8682 | 1.1297 | 1.6081 | 2.3690 | 0.4621 | 0.3263 |
| | $\xi = 0.8$ | 0.6946 | 1.0292 | 1.8083 | 3.2462 | 5.8413 | 0.7394 | 1.1404 |
| | $\xi = 0.0$ | 1.0000 | 1.0000 | 1.0000 | 1.0000 | 1.0000 | 0.0000 | 0.0000 |
| | $\xi = 0.1$ | 1.0001 | 1.0102 | 1.0304 | 1.0608 | 1.1017 | 0.1002 | 0.0100 |
| Uniform | $\xi = 0.2$ | 1.0001 | 1.0405 | 1.1210 | 1.2446 | 1.4170 | 0.2004 | 0.0400 |
| Distribution | $\xi = 0.35$ | 1.0003 | 1.1236 | 1.3699 | 1.7665 | 2.3674 | 0.3508 | 0.1232 |
| | $\xi = 0.5$ | 1.0004 | 1.2519 | 1.7545 | 2.6211 | 4.0781 | 0.5011 | 0.2517 |
| | $\xi = 0.8$ | – – – | – – – | – – – | – – – | – – – | – – – | – – – |

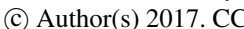
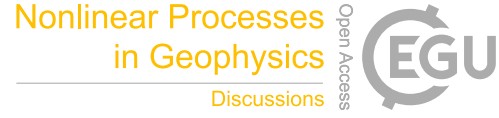


**Table 3.** Moments of ensemble averaged distributions (10000 ensembles) used for the one-dimensional chain (256 element long).

| Distribution | Disorder | $< M_1 >$ | $< M_2 >$ | $< M_3 >$ | $< M_4 >$ | $< M_5 >$ | $\Xi$ | $\Xi^2$ |
|---|---|---|---|---|---|---|---|---|
| | $\xi = 0.0$ | 1.0000 | 1.0000 | 1.0000 | 1.0000 | 1.0000 | 0.0000 | 0.0000 |
| | $\xi = 0.1$ | 1.0000 | 1.0100 | 1.0299 | 1.0601 | 1.1011 | 0.0999 | 0.0100 |
| Normal | $\xi = 0.2$ | 1.0000 | 1.0399 | 1.1196 | 1.2439 | 1.4225 | 0.1998 | 0.04 |
| Distribution | $\xi = 0.35$ | 1.0000 | 1.1192 | 1.3598 | 1.7648 | 2.4222 | 0.3456 | 0.1197 |
| | $\xi = 0.5$ | 1.0000 | 1.2093 | 1.6491 | 2.4617 | 3.9545 | 0.4579 | 0.2101 |
| | $\xi = 0.8$ | 1.0000 | 1.3319 | 2.0893 | 3.6833 | 7.1170 | 0.5767 | 0.3332 |
| | $\xi = 0.0$ | 1.0000 | 1.0000 | 1.0000 | 1.0000 | 1.0000 | 0.0000 | 0.0000 |
| | $\xi = 0.1$ | 1.0000 | 1.0100 | 1.0299 | 1.0599 | 1.1001 | 0.1000 | 0.0100 |
| Binary | $\xi = 0.2$ | 1.0000 | 1.0399 | 1.1196 | 1.2409 | 1.4069 | 0.2000 | 0.0400 |
| Distribution | $\xi = 0.35$ | 1.0000 | 1.1222 | 1.3668 | 1.7494 | 2.3006 | 0.3501 | 0.1226 |
| | $\xi = 0.5$ | 1.0000 | 1.2496 | 1.7502 | 2.5665 | 3.8279 | 0.5004 | 0.2506 |
| | $\xi = 0.8$ | 1.0000 | 1.6417 | 2.9340 | 5.3080 | 9.6373 | 0.8017 | 0.6442 |
| | $\xi = 0.0$ | 1.0000 | 1.0000 | 1.0000 | 1.0000 | 1.0000 | 0.0000 | 0.0000 |
| | $\xi = 0.1$ | 1.0000 | 1.0100 | 1.0299 | 1.0600 | 1.1006 | 0.1000 | 0.0100 |
| Uniform | $\xi = 0.2$ | 1.0000 | 1.0399 | 1.1197 | 1.2422 | 1.4134 | 0.2000 | 0.0400 |
| Distribution | $\xi = 0.35$ | 1.0000 | 1.1223 | 1.3670 | 1.7616 | 2.3605 | 0.3501 | 0.1227 |
| | $\xi = 0.5$ | 1.0000 | 1.2499 | 1.7507 | 2.6167 | 4.0775 | 0.5005 | 0.2509 |
| | $\xi = 0.8$ | 1.0000 | 1.6430 | 2.9350 | 5.6380 | 11.2983 | 0.8021 | 0.6455 |





**Table 4.** Scaled Coherent Wave Velocity Picking for different particles before and after localization length for a disordered chain with normal distribution (256 element long, 500 ensembles).

| Particle Number | Disorder | Average Mass | 5% Peak | 10% Peak | 70% Peak | 90% Peak | Peak | Zero Crossing |
|---|---|---|---|---|---|---|---|---|
| | $\xi = 0.0$ | 1.0000 | 1.0462 | 1.0365 | 1.0002 | 0.9911 | 0.9808 | 0.9560 |
| | $\xi = 0.1$ | 1.0000 | 1.0462 | 1.0365 | 1.0002 | 0.9911 | 0.9817 | 0.9560 |
| $130^{th}$ particle | $\xi = 0.2$ | 1.0000 | 1.0508 | 1.0409 | 1.0032 | 0.9938 | 0.9839 | 0.9560 |
| | $\xi = 0.35$ | 1.0000 | 1.0623 | 1.0515 | 1.0100 | 0.9994 | 0.9881 | 0.9525 |
| | $\xi = 0.5$ | 1.0000 | 1.0735 | 1.0616 | 1.0155 | 1.0035 | 0.9906 | 0.9449 |
| | $\xi = 0.8$ | 1.0000 | 1.0841 | 1.0713 | 1.0211 | 1.0079 | 0.9933 | 0.9328 |
| | $\xi = 0.0$ | 1.0000 | 1.0402 | 1.0317 | 0.9990 | 0.9910 | 0.9825 | 0.9597 |
| | $\xi = 0.1$ | 1.0000 | 1.0419 | 1.0332 | 1.0003 | 0.9920 | 0.9835 | 0.9599 |
| $150^{th}$ particle | $\xi = 0.2$ | 1.0000 | 1.0464 | 1.0373 | 1.0032 | 0.9946 | 0.9855 | 0.9597 |
| | $\xi = 0.35$ | 1.0000 | 1.0574 | 1.0475 | 1.0095 | 0.9998 | 0.9894 | 0.9566 |
| | $\xi = 0.5$ | 1.0000 | 1.0678 | 1.0569 | 1.0146 | 1.0036 | 0.9917 | 0.9500 |
| | $\xi = 0.8$ | 1.0000 | 1.0782 | 1.0664 | 1.0199 | 1.0076 | 0.9939 | 0.9387 |
| | $\xi = 0.0$ | 1.0000 | 1.0330 | 1.0258 | 0.9991 | 0.9924 | 0.9856 | 0.9665 |
| | $\xi = 0.1$ | 1.0000 | 1.0342 | 1.0271 | 1.0001 | 0.9933 | 0.9862 | 0.9666 |
| $200^{th}$ particle | $\xi = 0.2$ | 1.0000 | 1.0376 | 1.0303 | 1.0023 | 0.9954 | 0.9878 | 0.9665 |
| | $\xi = 0.35$ | 1.0000 | 1.0459 | 1.0380 | 1.0073 | 0.9995 | 0.9910 | 0.9642 |
| | $\xi = 0.5$ | 1.0000 | 1.0537 | 1.0450 | 1.0113 | 1.0025 | 0.9929 | 0.9587 |
| | $\xi = 0.8$ | 1.0000 | 1.0620 | 1.0526 | 1.0155 | 1.0056 | 0.9947 | 0.9494 |
| | $\xi = 0.0$ | 1.0000 | 1.0308 | 1.0242 | 0.9992 | 0.9930 | 0.9864 | 0.9685 |
| | $\xi = 0.1$ | 1.0000 | 1.0320 | 1.0253 | 1.0000 | 0.9937 | 0.9870 | 0.9686 |
| $220^{th}$ particle | $\xi = 0.2$ | 1.0000 | 1.0350 | 1.0282 | 1.0020 | 0.9954 | 0.9884 | 0.9685 |
| | $\xi = 0.35$ | 1.0000 | 1.0426 | 1.0352 | 1.0066 | 0.9993 | 0.9914 | 0.9665 |
| | $\xi = 0.5$ | 1.0000 | 1.0500 | 1.0419 | 1.0105 | 1.0022 | 0.9933 | 0.9619 |
| | $\xi = 0.8$ | 1.0000 | 1.0575 | 1.0487 | 1.0142 | 1.0050 | 0.9949 | 0.9542 |





**Table 5.** Unscaled Coherent Wave Velocity Picking ($\sqrt{\widetilde{M_1}}$) for different particles before and after localization length for a disordered chain with normal distribution (256 element long, 500 ensembles).

| Particle Number | Disorder | Average Mass | 5% Peak | 10% Peak | 70% Peak | 90% Peak | Peak | Zero Crossing |
|---|---|---|---|---|---|---|---|---|
| | $\xi = 0.1$ | 1.0000 | 1.0462 | 1.0366 | 1.0002 | 0.9912 | 0.9818 | 0.9560 |
| | $\xi = 0.2$ | 1.0000 | 1.0509 | 1.0409 | 1.0033 | 0.9939 | 0.9840 | 0.9560 |
| $130^{th}$ particle | $\xi = 0.35$ | 1.0022 | 1.0614 | 1.0505 | 1.0091 | 0.9985 | 0.9872 | 0.9515 |
| | $\xi = 0.5$ | 1.0274 | 1.0595 | 1.0477 | 1.0022 | 0.9904 | 0.9776 | 0.9322 |
| | $\xi = 0.8$ | 1.1581 | 1.0081 | 0.9962 | 0.9496 | 0.9373 | 0.9237 | 0.8668 |
| | $\xi = 0.1$ | 1.0000 | 1.0420 | 1.0332 | 1.0003 | 0.9921 | 0.9835 | 0.9599 |
| | $\xi = 0.2$ | 1.0000 | 1.0465 | 1.0374 | 1.0032 | 0.9946 | 0.9856 | 0.9597 |
| $150^{th}$ particle | $\xi = 0.35$ | 1.0022 | 1.0564 | 1.0465 | 1.0086 | 0.9989 | 0.9885 | 0.9556 |
| | $\xi = 0.5$ | 1.0274 | 1.0539 | 1.0431 | 1.0013 | 0.9905 | 0.9787 | 0.9373 |
| | $\xi = 0.8$ | 1.1581 | 1.0026 | 0.9917 | 0.9485 | 0.9370 | 0.9243 | 0.8723 |
| | $\xi = 0.1$ | 1.0000 | 1.0343 | 1.0271 | 1.0001 | 0.9934 | 0.9862 | 0.9666 |
| | $\xi = 0.2$ | 1.0000 | 1.0377 | 1.0304 | 1.0024 | 0.9954 | 0.9879 | 0.9665 |
| $200^{th}$ particle | $\xi = 0.35$ | 1.0022 | 1.0449 | 1.0370 | 1.0064 | 0.9985 | 0.9901 | 0.9631 |
| | $\xi = 0.5$ | 1.0274 | 1.0399 | 1.0313 | 0.9981 | 0.9894 | 0.9799 | 0.9458 |
| | $\xi = 0.8$ | 1.1581 | 0.9876 | 0.9788 | 0.9443 | 0.9351 | 0.9250 | 0.8822 |
| | $\xi = 0.1$ | 1.0000 | 1.0320 | 1.0253 | 1.0000 | 0.9937 | 0.9870 | 0.9686 |
| | $\xi = 0.2$ | 1.0000 | 1.0320 | 1.0253 | 1.0000 | 0.9937 | 0.9870 | 0.9685 |
| $220^{th}$ particle | $\xi = 0.35$ | 1.0022 | 1.0417 | 1.0343 | 1.0057 | 0.9984 | 0.9905 | 0.9654 |
| | $\xi = 0.5$ | 1.0274 | 1.0362 | 1.0283 | 0.9972 | 0.9891 | 0.9803 | 0.9490 |
| | $\xi = 0.8$ | 1.1581 | 0.9834 | 0.9752 | 0.9431 | 0.9346 | 0.9252 | 0.8867 |





**Table 6.** Coherent Wave Velocity calculated from the time taken by the pulse to travel a common distance of separation (7 particles/elements) with time calculated in reference to $5\%, 10\%, 70\%, 90\%$ of the peak value and the peak value of the coherent wave packet.

| Particle Number | Disorder | Average Mass | 5% Peak | 10% Peak | 70% Peak | 90% Peak | Peak | Zero Crossing |
|---|---|---|---|---|---|---|---|---|
| | $\xi = 0.0$ | 1.0000 | 1.0135 | 1.0089 | 0.9999 | 0.9954 | 0.9954 | 0.9867 |
| | $\xi = 0.1$ | 1.0000 | 1.0143 | 1.0108 | 1.0000 | 0.9970 | 0.9932 | 0.9867 |
| $137^{th}$ particle | $\xi = 0.2$ | 1.0000 | 1.0167 | 1.0136 | 1.0010 | 0.9980 | 0.9941 | 0.9867 |
| - $130^{th}$ | $\xi = 0.35$ | 1.0000 | 1.0227 | 1.0186 | 1.0039 | 0.9996 | 0.9951 | 0.9910 |
| particle | $\xi = 0.5$ | 1.0000 | 1.0284 | 1.0241 | 1.0058 | 1.0006 | 0.9945 | 0.9867 |
| | $\xi = 0.8$ | 1.0000 | 1.0319 | 1.0279 | 1.0129 | 1.0088 | 1.0036 | 0.9867 |
| | $\xi = 0.0$ | 1.0000 | 1.0135 | 1.0089 | 0.9999 | 0.9954 | 0.9954 | 0.9867 |
| | $\xi = 0.1$ | 1.0000 | 1.0107 | 1.0082 | 0.9982 | 0.9960 | 0.9930 | 0.9867 |
| $157^{th}$ particle | $\xi = 0.2$ | 1.0000 | 1.0105 | 1.0080 | 0.9974 | 0.9957 | 0.9928 | 0.9867 |
| - $150^{th}$ | $\xi = 0.35$ | 1.0000 | 1.0072 | 1.0054 | 0.9984 | 0.9965 | 0.9948 | 0.9910 |
| particle | $\xi = 0.5$ | 1.0000 | 1.0056 | 1.0041 | 0.9983 | 0.9964 | 0.9940 | 0.9867 |
| | $\xi = 0.8$ | 1.0000 | 1.0106 | 1.0072 | 0.9947 | 0.9917 | 0.9880 | 0.9780 |
| | $\xi = 0.0$ | 1.0000 | 1.0135 | 1.0089 | 0.9999 | 0.9954 | 0.9954 | 0.9910 |
| | $\xi = 0.1$ | 1.0000 | 1.0090 | 1.0073 | 0.9969 | 0.9959 | 0.9948 | 0.9867 |
| $227^{th}$ particle | $\xi = 0.2$ | 1.0000 | 1.0074 | 1.0056 | 0.9968 | 0.9951 | 0.9934 | 0.9867 |
| - $220^{th}$ | $\xi = 0.35$ | 1.0000 | 1.0056 | 1.0039 | 0.9957 | 0.9939 | 0.9917 | 0.9867 |
| particle | $\xi = 0.5$ | 1.0000 | 1.0059 | 1.0039 | 0.9968 | 0.9950 | 0.9926 | 0.9867 |
| | $\xi = 0.8$ | 1.0000 | 1.0122 | 1.0111 | 1.0062 | 1.0049 | 1.0031 | 0.9867 |
| | $\xi = 0.0$ | 1.0000 | 1.0089 | 1.0089 | 0.9999 | 0.9999 | 0.9954 | 0.9910 |
| | $\xi = 0.1$ | 1.0000 | 1.0087 | 1.0060 | 0.9975 | 0.9966 | 0.9953 | 0.9910 |
| $247^{th}$ particle | $\xi = 0.2$ | 1.0000 | 1.0073 | 1.0047 | 0.9964 | 0.9952 | 0.9925 | 0.9910 |
| - $240^{th}$ | $\xi = 0.35$ | 1.0000 | 1.0041 | 1.0019 | 0.9946 | 0.9928 | 0.9909 | 0.9954 |
| particle | $\xi = 0.5$ | 1.0000 | 1.0017 | 1.0002 | 0.9928 | 0.9916 | 0.9903 | $----$ |
| | $\xi = 0.8$ | 1.0000 | 0.9937 | 0.9919 | 0.9860 | 0.9846 | 0.9847 | $----$ |