# Peer review of "Effect of Disorder on Bulk Sound Wave Speed : A Multiscale Spectral Analysis."

_Nonlinear Processes in Geophysics, 2016_

## Referee Comment (RC1) · Anonymous Referee #1 · 28 Feb 2017

Review of NPG-2016-83, by Rohit Kumar Shrivastava and Stefan Luding, "Effect of Disorder on Bulk SoundWave Speed : A Multiscale Spectral Analysis"

The authors have presented interesting results of a numerical study on wave propagation through a precompressed mass disordered granular chain. The paper is reasonably well written and is worth to be published.

Typographical mistakes and other minor corrections 1. Page 11 string 9 ". . .the shape of the distribution (h)as no effect. . ." 2. Page 17 string 8 Figure ?? 3. Page 20 string 13 ". . .far away from the source (Fig (probably, Eq.) 41 and Table 6)."

Please also note the supplement to this comment:

[Figure]

http://www.nonlin-processes-geophys-discuss.net/npg-2016-83/npg-2016-83-RC1-supplement.pdf
* * *

---

## Referee Comment (RC2) · Anonymous Referee #2 · 12 Mar 2017

The manuscript aims to report the results of investigation of wave propagation in a 1D chain of particles with randomly assigned masses. The particles are connected by longitudinal springs with power law dependence between the force and displacement. The exponent is assumed either 0 (conventional linear spring) or 0.5 (Herzian contact). The manuscript however considers only linear contacts (exponent=0). The way it is implemented cannot be considered as a linearization of Herzian contact for small displacement, since to do so one needs to specify the point around which linearization is performed and then linearize the force-displacement law around it. As a result the linearized law will contain the exponent as a parameter. This is shown in eq. (8), which is linearization by itself. Then a parametric analysis of the influence of

the exponent needs to be performed. Otherwise, the claim about non-linearity is not justified. In the simple linear case such an investigation could have already existed in the literature, as the lit review does not make it clear. If the non-linearity focus of the paper is kept one needs more thorough literature review. In particular, discussing the cases when linearization cannot be performed as in the case of Herzian contact with zero displacement or the case of bilinear springs (see e.g., Dyskin, A.V., E. Pasternak and E. Pelinovsky, 2012. Periodic motions and resonances of impact oscillators. Journal of Sound and Vibration 331(12) 2856-2873; Dyskin, A.V., E. Pasternak and I. Shufrin, 2014. Structure of resonances and formation of stationary points in symmetrical chains of bilinear oscillators. Journal of Sound and Vibration 333, 6590–6606; Guzek, A., A.V. Dyskin, E. Pasternak and I. Shufrin, 2016. Asymptotic analysis of bilinear oscillators with preload. International Journal of Engineering Science, 106, 125-141 and the literature cited there). The literature review needs to be expanded, as the other components of displacements and rotations are barely mentioned. The review should include more papers where dynamics and wave propagation in particle sets and chains is investigated (see e.g., Pasternak, E. and Mühlhaus, H.-B. (2005) Generalised homogenisation procedures for granular materials, Eng Math, 52, Number 1, 199-229; Dyskin, A.V., E. Pasternak and G. Sevel, 2014. Chains of oscillators with negative stiffness elements. Journal of Sound and Vibration, 333, Issue 24, 6676–6687; Esin, M., Pasternak, E. and Dyskin, A.V. (2016) Stability of 2D discrete mass-spring systems with negative stiffness springs, Physica Status Solidi (B) Basic Research, 253, 7, 1395-1409 and the literature cited there). It is not clear from the text whether the disorder manifests itself through the non-uniformity of masses only or non-uniformity of the inter-particle distance is included as well. The measures of disorder adopted in the paper need to be explained in more detail. Figures, especially Figs. 4-6 need to be made larger. Caption to Fig. 5 is not informative and should be rewritten. Abstract: the references should be removed or shortened. English needs to be improved, the use of articles checked. Word "media" is plural; with article "a" should go singular, "medium". I suggest major revision to address the above comments.

---

## Author Comment (AC1) · 7 May 2017

**1   Anonymous Reviewer 1**

**1.0.1   Reviewer's Comment 1**

The authors have presented interesting results of a numerical study on wave propagation through a precompressed mass disordered granular chain. The paper is reasonably well written and is worth to be published.

**1.0.2 Author's Response**

The authors would like to thank the reviewer for the motivational comments and the recommendation for the paper to be published.

**1.0.3 Reviewer's Comment 2**

Typographical mistakes and other minor corrections
1. Page 11 string 9 ". . .the shape of the distribution (h)as no effect. . ."
2. Page 17 string 8 Figure ??
3. Page 20 string 13 ". . .far away from the source (Fig (probably, Eq.) 41 and Table 6)."

**1.0.4 Author's Response**

Typographical mistakes have been taken into account and corrections have been made as highlighted by the reviewer.

**1.0.5 Author's changes in manuscript**

1. ". . .the shape of the distribution has no effect. . ."
2. ". . .Figure 10 shows the participation. . ."
3. ". . .far away from the source (Fig. 6 and Table 6). The group velocity given by Eq. (45) also shows decrease in wave-speed with increase in disorder."

---

## Author Comment (AC2) · 11 Jun 2017

**0.0.1 Referee's Comment 1**

The manuscript aims to report the results of investigation of wave propagation in a 1D chain of particles with randomly assigned masses. The particles are connected by longitudinal springs with power law dependence between the force and displacement. The exponent is assumed either 0 (conventional linear spring) or 0.5 (Hertzian contact). The manuscript however considers only linear contacts (exponent=0). The way it is implemented cannot be considered as a linearization of Hertzian contact for small

displacement, since to do so one needs to specify the point around which linearization is performed and then linearize the force-displacement law around it. As a result the linearized law will contain the exponent as a parameter. This is shown in eq. (8), which is linearization by itself. Then a parametric analysis of the influence of the exponent needs to be performed. Otherwise, the claim about non-linearity is not justified.

**0.0.2 Author's Response**

The authors would like to thank the Referee for highlighting the inconsistency of the linearization done. However, about the Referee's comment that " The way it is implemented cannot be considered as a linearization of Hertzian contact for small displacement ", we disagree. The characteristic overlap of particles ($\Delta_{(i,j)}$) is the initial configuration around which linearization is performed as shown in Eq. (10) (present manuscript) of the revised manuscript. On using Eq. (2) and (4), Eq. (11) can be written as Eq. (14) which is a linearized version of the nonlinear equation of motion (Eq. (8)). Since, we are concerned with only mass disorder, we choose $\kappa = 1$ (Lawney and Luding (2014)). We are concerned with only mass disorder because it has larger contribution towards disorder than stiffness (highlighted previously in Achilleos et al. (2016)); this proposition has been mentioned in the Introduction section of the manuscript. The removal of factor $\frac{1}{1+\beta}$ has also been discussed in Sect. 2.3 ( $\beta = 1/2$ for Hertzian repulsive interaction force). Section 2.2 and the results associated with Sect. 2.2 have been added in Sect. 4.1.2 for comparing the results (space time responses). The space time responses have been computed numerically from a nonlinear equation of motion (Eq. (8); Hertzian repulsive interaction force) and from an analytical solution (used for further analyses in the manuscript) of a linear equation of motion (Eq. (14)), the space time responses obtained for both the linear and nonlinear equations coincide (Fig. 3) indicating that the solution from the linearized equation of motion can be used for further analyses.
**0.0.3 Author's changes in manuscript**

Section 2.2 has been replaced by Sect. 2.2 and 2.3. Section 4.1.2 has been added with Fig. 3 and Fig. 4.

**0.0.4 Referee's Comment 2**

In the simple linear case such an investigation could have already existed in the literature, as the lit review does not make it clear. If the non-linearity focus of the paper is kept one needs more thorough literature review. In particular, discussing the cases when linearization cannot be performed as in the case of Herzian contact with zero displacement or the case of bilinear springs (see e.g., Dyskin, A.V., E. Pasternak and E. Pelinovsky, 2012. Periodic motions and resonances of impact oscillators. Journal of Sound and Vibration 331(12) 2856-2873; Dyskin, A.V., E. Pasternak and I. Shufrin, 2014. Structure of resonances and formation of stationary points in symmetrical chains of bilinear oscillators. Journal of Sound and Vibration 333, 6590–6606; Guzek, A., A.V. Dyskin, E. Pasternak and I. Shufrin, 2016. Asymptotic analysis of bilinear oscillators with preload. International Journal of Engineering Science, 106, 125-141 and the literature cited there). The literature review needs to be expanded, as the other components of displacements and rotations are barely mentioned. The review should include more papers where dynamics and wave propagation in particle sets and chains is investigated (see e.g., Pasternak, E. and Mühlhaus, H.-B. (2005) Generalised homogenisation procedures for granular materials, Eng Math, 52, Number 1, 199-229; Dyskin, A.V., E. Pasternak and G. Sevel, 2014. Chains of oscillators with negative stiffness elements. Journal of Sound and Vibration, 333, Issue 24, 6676–6687; Esin, M., Pasternak, E. and Dyskin, A.V. (2016) Stability of 2D discrete mass-spring systems with negative stiffness springs, Physica Status Solidi (B) Basic Research, 253, 7, 1395-1409 and the literature cited there).

**0.0.5 Author's Response**

The literature review has been expanded and over 55 references have been cited in total including a few very recent ones. The referee's recommendations have also been taken into account and cited in the manuscript. The Introduction section comprises much more relevant citations for highlighting the importance of the research work and the previous investigations done related to the ones done in this manuscript have been duly cited. The mathematical models which have been used previously are now better cited in the respective sections.

**0.0.6 Author's changes in the manuscript**

Changes have been made in the Introduction section as well as in the respective sections related to the relevant references.

**0.0.7 Referee's Comment 3**

It is not clear from the text whether the disorder manifests itself through the non-uniformity of masses only or non-uniformity of the inter-particle distance is included as well. The measures of disorder adopted in the paper need to be explained in more detail.

**0.0.8 Author's Response**

The disorder in granular materials manifests as a result of the granules/particles, we focus on mass disorder alone. The granular/particular properties of the material like size and shape are instrumental in characterizing the bulk disorder of the materials. The initial stress/pressure (initial configuration), in our case characteristic overlap $(\Delta_{(i,j)})$

determines the inter-particle distance in conjunction with the size of the granules. It can be inferred from Sect. 2 that small impulse magnitude ($v_o$) causes small displacement of particles from the initial position, the amplitude of vibration is less, the pristine nature of initial configuration is not disturbed (no opening and closing of contacts; no shock waves, only sound waves). Hence, the disorder manifests itself only through the non-uniformity of masses and contact stiffness. However, as mentioned in the Introduction section that " mass disorder has much stronger contribution towards disorder ", mass disorder has been chosen for further analyses.

Details regarding disorder has been mentioned in Sect. 2.6. The standard deviation of the mass distribution has been chosen as the disorder parameter ($\xi$). For comparisons between different distributions viz. Normal, Binary and Uniform distributions, scaled moments have been utilized which have been mentioned in detail in Appendix C. The reason behind the choice of these distribution was to p-reserve symmetry around mean mass (first scaled moment), at least for not too large $\xi$.

**0.0.9 Author's changes in the manuscript**

"Equal size" has been added to abstract.

**0.0.10 Referee's Comment 4**

Figures, especially Figs. 4-6 need to be made larger. Caption to Fig. 5 is not informative and should be rewritten.

**0.0.11 Author's Response**

Figures are complying with the protocol fixed by the journal - we thought. However, the figures are of high quality.

**0.0.12 Author's changes in manuscript**

The labels and legends have been enhanced for viewers' comfort. The plots have also been enhanced. Caption to Fig. 7 (present manuscript) has been rewritten and is now more informative; if there are still problems, please specify.

**0.0.13 Referee's Comment 5**

Abstract: the references should be removed or shortened.

**0.0.14 Author's Response**

The authors agree with the Referee's comment, the references have been mentioned in the introduction section and need not be mentioned in the abstract.

**0.0.15 Author's changes in manuscript**

The references have been removed from the abstract.

**0.0.16 Referee's Comment 6**

English needs to be improved, the use of articles checked. Word "media" is plural; with article "a" should go singular, "medium".

**0.0.17 Author's Response**

The spelling of words and the grammar of English used in the manuscript have been checked. Special attention has also been given to the use of articles, especially when the word "media" has been used, the article "a" has been removed as instructed by the referee.

**0.0.18 Author's changes in manuscript**

: English has been improved throughout.

**0.0.19 Referee's Comment 7**

I suggest major revision to address the above comments.

**0.0.20 Author's Response**

Major revisions have been done in the manuscript to address the comments of the Referee. The authors hope that the revised manuscript meets the expectations of the Referee.

Please also note the supplement to this comment:
http://www.nonlin-processes-geophys-discuss.net/npg-2016-83/npg-2016-83-AC2-supplement.pdf

[Figure]

**Fig. 1.** Figure 3 of the revised manuscript.

[Figure]

**Fig. 2.** Figure 4 of the revised manuscript.